# The gut ileal mucosal virome is disturbed in patients with Crohn's disease and exacerbates intestinal inflammation in mice

Zhirui Cao[1,2,3,4,13], Dejun Fan[2,5,6,13], Yang Sun[7,8,9,13] ✉, Ziyu Huang[1,2,3,4], Yue Li[1,2,3,4], Runping Su[1,2,3,4], Feng Zhang[1,2,3,4], Qing Li[10], Hongju Yang[7,9], Fen Zhang[11], Yinglei Miao ●[7,8], Ping Lan ●[1,2,3,4,6,12], Xiaojian Wu[1,2,3,4,6,12] ✉ & Tao Zuo ●[1,2,3,4] ✉

Gut bacteriome dysbiosis is known to be implicated in the pathogenesis of inflammatory bowel disease (IBD). Crohn's disease (CD) is an IBD subtype with extensive mucosal inflammation, yet the mucosal virome, an empirical modulator of the bacteriome and mucosal immunity, remains largely unclear regarding its composition and role. Here, we exploited trans-cohort CD patients and healthy individuals to compositionally and functionally investigate the small bowel (terminal ileum) virome and bacteriome. The CD ileal virome was characterised by an under-representation of both lytic and temperate bacteriophages (especially those targeting bacterial pathogens), particularly in patients with flare-up. Meanwhile, the virome-bacteriome ecology in CD ileal mucosa was featured by a lack of *Bifidobacterium*- and Lachnospiraceae-led mutualistic interactions between bacteria and bacteriophages; surprisingly it was more pronounced in CD remission than flare-up, underlining the refractory and recurrent nature of mucosal inflammation in CD. Lastly, we substantiated that ileal virions from CD patients causally exacerbated intestinal inflammation in IBD mouse models, by reshaping a gut virome-bacteriome ecology preceding intestinal inflammation (microbial trigger) and augmenting microbial sensing/defence pathways in the intestine cells (host response). Altogether, our results highlight the significance of mucosal virome in CD pathogenesis and importance of mucosal virome restoration in CD therapeutics.

Inflammatory bowel disease (IBD) is a chronic autoimmune disease of the gastrointestinal (GI) tract, characterised by a progressive process involving intermittent flare-ups and remissions of intestinal inflammation[1]. Crohn's Disease (CD) and Ulcerative Colitis (UC) are two major subtypes of IBD afflicting disparate regions of the GI tract. While UC afflicts the colon, CD occurs anywhere in the GI tract, particularly the ileum, which results in a more complicated, severe phenotype in disease manifestation and course[1]. Although the exact cause of CD is

still unclear, the disturbed gut microbiome was demonstrated to underpin the aetiology and pathogenesis of CD[2,3]. However, most of the existing studies in CD microbiome were based on the faecal bacteriome, and little is known about the virome alteration at the intestinal mucosal level or its causality/pathogenicity in CD inflammation.

As an important microbial inhabitant in the gut, the critical role of gut virome in human health has been appreciated in recent years[4]. The human faecal virome is composed of bacteriophages and eukaryotic

viruses, with a predominance of DNA viruses[5]. However, the overall composition and function of the gut virome at the intestinal mucosa (particularly in small bowel) are not depicted in humans to date. Due to the close proximity of viruses/bacteria to the intestinal mucosa, mucosa-associated microbes are poised to play a more pronounced role than their faecal counterparts in host physiology and pathophysiology. Anecdotal studies in mice and in vitro have reported that individual bacteriophages can play both beneficial and detrimental roles in gut homoeostasis and inflammation[6–8]. Mucus-adherent T4 phage can express Immunoglobulin (Ig)-like protein domains on their capsid which bind to mucin glycoproteins at the intestinal mucus layer, therefore defending against infection of the intestinal epithelium by *Escherichia coli*[6,7]. By contrast, administration of *Escherichia* bacteriophages exacerbated colonic inflammation in mice with colitis[8]. Nonetheless, it remains largely unknown about how mucosal virome as a community regulates intestinal inflammation in humans. The ileum, the terminal part of the small bowel, is the primary site of CD infliction where various obligate and facultative anaerobic bacteria reside[9]. Meanwhile, CD is characterised by an overt intestinal bacteriome dysbiosis and sophisticated involvement of intestinal wall (transmural) inflammation[10]. Being cognizant of the nature of bacteriophages preying on bacteria and of viruses orchestrating host immunity, we herein postulated that mucosal virome disturbance might be a key component underlying intestinal inflammation in CD.

CD is a multi-factorial, highly heterogeneous disease that is intricately linked to host-extrinsic factors (population geography, diet, drugs, and lifestyle) and host-intrinsic factors (disease course and phenotype)[10]. These factors are also prominent factors in defining the human gut bacteriome[11,12], it is therefore desired to tease apart the impacts of these factors on the mucosal virome in relation to CD pathogenesis. Given that CD stems from mucosal barrier dysfunction and that bacteriophages/viruses may play a primordial role in this process, it is of paramount importance to understand the mucosal virome in CD. In this work, we elucidate the composition and functionality of the gut virome (comprising prokaryotic and eukaryotic virome) at the intestinal mucosal level in both CD and health, and observe its associations with clinical factors. We substantiate that ileal mucosal virions from CD patients can causally exacerbate intestinal inflammation in IBD mouse models, mechanistically through reshaping the bacteriome-virome ecology, towards a pro-inflammatory enterotype. Our results highlight that restoring the disrupted virome-bacteriome network at the intestinal mucosa should be considered a novel goal in CD therapeutics in the future.

## Result

### The landscape of ileal virome in CD and healthy individuals

To investigate the small bowel virome configuration, we enrolled a total of 208 participants from two separate cohorts in China, Guangzhou (*n* = 102) and Kunming (*n* = 106) (Fig. 1a), and collected terminal ileum (TI) biopsies from each subject (Summarised in Table 1 for each cohort; metadata for individual participants were detailed in Supplementary Data 1). Patients with CD were stratified into remission or flare-up based on the clinical assessment of Harvey–Bradshaw index (HBI, a simpler version of CD activity index): 41.6% of patients were in clinical remission (defined as HBI < 5), while 58.4% of patients were in flare-up (defined as HBI ≥ 5). We then conducted viral-like particles (VLPs) enrichment from the TI mucosa, followed by ultra-deep virome sequencing and profiling (Fig. 1b). For simultaneous bacteriome interrogation, we also performed 16S rDNA sequencing and bacteriome profiling on the same original mucosal specimen. Consequently, we obtained 21.8 ± 4.2 million (mean ± S.E.) pair-end clean reads per sample from virome sequencing, and 82.2 ± 10.8 thousand pair-end clean reads per sample from 16S rDNA sequencing. In parallel to mucosal sampling, 99.52% (*n* = 207) of participants completed the questionnaire survey consisting of 46 metadata variables, ranging

from disease phenotype (documented by physicians), medications (retrieved through medical records), geographic regions, anthropometrics (self-reported by participants), and dietary habits in the past 3 months (self-reported by participants under the guidance of a dietitian). These trans-regional cohorts and the comprehensive metadata profile enabled us to investigate the alterations in ileal mucosal virome and to identify the clinical covariates in CD.

We first explored the community structure landscape of the ileal virome in HC and CD. Overall, the composition profile of ileal virome from all study participants formed a continuum, exhibiting substantial heterogeneity across individuals, at both the order and family levels (Fig. 1c and Supplementary Fig. 2a, b). Moreover, the compositional heterogeneity was significantly higher in CD than in HC (Bartlett's test of homogeneity of variances, *p* = 0.022, Fig. 1f). The ileal virome was dominated by the orders *Petitvirales*, unclassified *Caudoviricetes*, *Crassvirales*, and *Crassvirales* (Supplementary Fig. 2a). It is worth noting that the viral order *Caudovirales* was recently disbanded by the International Committee on Taxonomy of Viruses (ICTV) and reclassified into new taxonomy groups including the *Crassvirales* order, and the *Ackermannviridae*, *Chaseviridae* and *Herelleviridae* families[13]; we therefore adhered to the new viral taxonomy classification criteria and did not report *Caudovirales* which was previously reported to be the most abundant order in the human faecal virome[5]. At the family level, the ileal mucosal virome was dominated by *Microviridae*, unclassified *Caudoviricetes* family, *Circoviridae*, and *Anelloviridae*, collectively accounting for >80% relative abundance in 74.5% of participants (Fig. 1c, d).

While the majority of the viral orders and families in the ileal virome were prokaryotic viruses, we sought to assess the presence of eukaryotic viruses and their ratios with prokaryotic viruses at the ileal mucosa across the study participants. The configuration of the eukaryotic virome was also highly individualised across all study participants in the two cohorts (Supplementary Fig. 2c, d), with the relative abundance of eukaryotic viruses ranging from 0.24% to 95.68% (Median: 26.81%, Supplementary Fig. 2e). Correspondingly, prokaryotic viruses constituted a substantially larger proportion of the mucosal virome, making up to 73.19% of the total virome (Supplementary Fig. 2e), dominated by lytic phages with a median percentage of 78.39% (Fig. 1e). Taken together, the gut mucosal virome at the ileal mucosa site of the GI tract is predominated by lytic phages and is greatly heterogeneous across individuals.

### Intestinal inflammation and core host factors associated with ileal virome composition

Building upon the clinical metadata and the ileal microbiome composition profile of all participants, we probed the impact of 46 metadata factors on the variations of ileal viromes and bacteriomes across participants. Of these factors, 9 factors were identified to significantly associate with the ileal virome composition, including intestinal inflammation (CD remission versus flare-up versus non-inflammation), CD versus HC, diet (including consumption of alcoholic beverages, coffee, and puffed food), medications (including biologics, immunosuppressants, and glucocorticoids), and geographic region (FDR-adjusted *p* < 0.05, Fig. 1g). Among them, intestinal inflammation had a stronger explanatory power on the ileal virome variation compared to the mere effect size of the CD versus HC (Fig. 1g), indicating that the inflammation course of CD further influenced the ileal virome composition. Combined, the cumulative effect size of all host factors was 23.58% (Fig. 1h). We then grouped the 46 host factors into predefined 7 categories, and assessed the combined effect size of each category. Again, mucosal inflammation in CD showed the largest effect on the ileal virome composition, accounting for 1.5% of the ileal virome variations, while geography, medications, and dietary habit also showed a significant impact in descending order of effect size (all FDR-adjusted *p* < 0.05, Supplementary Fig. 3a). These data suggest that while

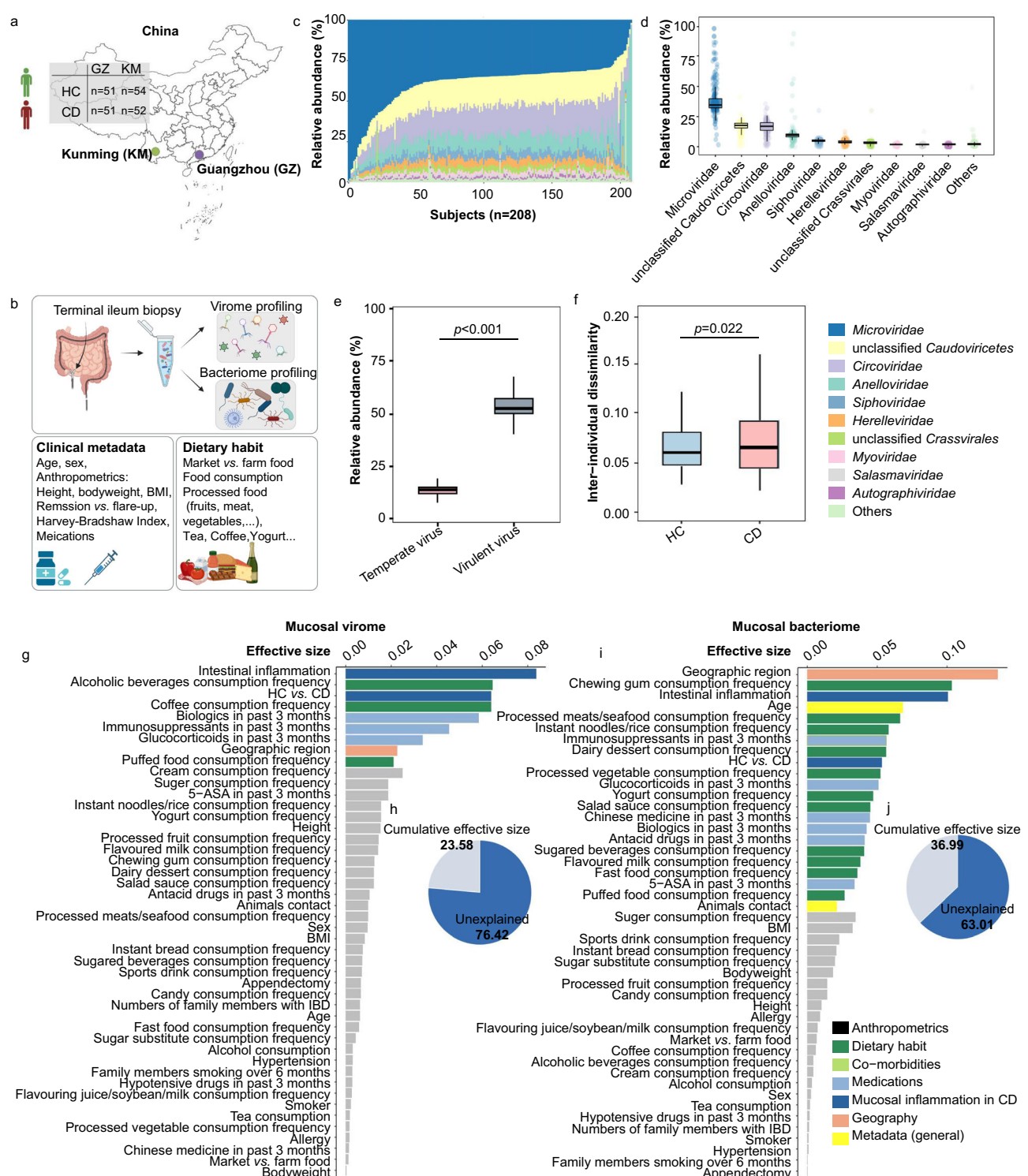

**g** Mucosal virome

**i** Mucosal bacteriome

**h** Cumulative effective size

**j** Cumulative effective size

Anthropometrics
Dietary habit
Co-morbidities
Medications
Mucosal inflammation in CD
Geography
Metadata (general)

intestinal inflammation is the primary factor influencing ileal virome composition, other host factors are core covariates of ileal virome.

In contrast, geography was shown to have the largest effect size on the ileal bacteriome variations (Fig. 1i and Supplementary Fig. 3b), followed by medications, dietary habit, and mucosal inflammation in CD (Supplementary Fig. 3b). Analogous to our prior study investigating the faecal bacteriome and virome variations in healthy individuals[11], whilst geography was the most prominent factor in shaping the gut bacteriome, the gut (both faecal and mucosal) virome was more influenced by host (patho)physiology, particularly on the mucosal level (Fig. 1g, i). When comparing the host factors associated with the

ileal virome composition versus those associated with the ileal bacteriome composition, we found that 13 more factors were influential in impacting the ileal bacteriome (including consumption of processed foods, flavoured milk, antacid drugs, 5-aminosalicylic acid [5-ASA]) (Fig. 1g, i). Interestingly, these additional factors that associated with the bacteriome composition were classified into dietary components or drugs, which may serve as substrates and/or modulators for bacteria rather than for viruses. These findings were also validated by multi-omics analysis by MOFA[14], which identified 6 predominant microbiome factors that captured the most varying characteristics within the multi-omics mucosal microbiome dataset (differentially contributed

**Fig. 1 | Schematic overview of the study design, variations of the ileal mucosal virome in CD and healthy controls, and covariates of the ileal mucosal virome and bacteriome. a** Subject recruitment and cohort description. A total of 208 individuals including patients with Crohn's disease (CD) and healthy controls (HC) were recruited from two China sites, Guangzhou (GZ, HC: $n = 51$; CD: $n = 51$) and Kunming (KM, HC: $n = 54$; CD: $n = 52$). **b** Overview of the study design and data collection regime, including respective profiling of the ileal virome and bacteriome, clinical metadata survey, and record of dietary habits for each individual. **c** Variations of the ileal virome composition at the family level across all participants, plotted according to the ranking order of the relative abundance of each viral family. **d** The relative abundances of viral families in the ileal mucosal virome. **e** Percentage of temperate phages and lytic phages in the ileal mucosal virome. Statistical significance was determined by *t*-test, with 208 independent samples.

**f** Inter-individual Bray–Curtis dissimilarities between ileal virome in the CD and HC groups respectively. Statistical test was performed by Bartlett's test of homogeneity of variances, with 208 independent samples. **g** The effect size of metadata factors on human ileal virome variation. Virome covariates were identified via *envfit* (vegan) and those with statistical significance measured by a two-tailed permutation test (FDR adjusted $p < 0.05$) were coloured based on the predefined metadata categories (also shown in Supplementary Fig. 3a, b). Insignificant metadata factors were plotted in grey. **h** Pie chart shows the fraction of ileal virome variation explained by the interrogated metadata factors. **i** The effect size of metadata factors on human ileal bacteriome variation. **j** Pie chart shows the fraction of bacteriome variation explained by the interrogated metadata factors. For box plots, the boxes extend from the 1st to the 3rd quartile (25th to 75th percentiles), with the median depicted by a horizontal line.

## Table 1 | Subject and sample characteristics

| Cohort | Guangzhou (Guangdong Province) | | Kunming (Yunnan Province) | |
|---|---|---|---|---|
| Participants | HC | CD | HC | CD |
| Number (percentage) | 51 (24.52%) | 51 (24.52%) | 54 (25.96%) | 52 (25%) |
| Age (years) | 28.6 ± 3.97 | 29.6 ± 7.98 | 42.8 ± 9.48 | 38.1 ± 14.9 |
| BMI | 22.54 ± 2.86 | 20.37 ± 4.03 | 22.14 ± 2.08 | 20.73 ± 3.79 |
| Male | 30 | 38 | 35 | 23 |
| Female | 21 | 13 | 19 | 29 |
| Clinical treatment for CD | – | 46 | – | 31 |
| Glucocorticoids | – | 20 | – | 2 |
| 5-ASA | – | 27 | – | 22 |
| Immunosuppressants | – | 25 | – | 8 |
| Biologics | – | 32 | – | 8 |

Age and BMI are presented as mean ± standard deviation (BMI, body mass index).

by the virome and bacteriome variances, shown in Supplementary Fig. 3c; further detailed contributions by prokaryotic and eukaryotic virome were shown in Supplementary Fig. 3e, f), and then pinpointed the associations of patients' metadata with these microbiome factors (Supplementary Fig. 3d). The results showed that the microbiome factors 2 and 4, which captured microbial variations majorly sourced from virome, were significantly associated with CD-related phenotype (Supplementary Fig. 3d). In contrast, the microbiome factor 1, which predominantly captured microbial variations majorly sourced from bacteriome, had robust associations with patient geography and dietary habits-related factors (Supplementary Fig. 3d). Altogether, these data suggest that mucosal bacteriome may be largely influenced by host-extrinsic factors (such as geography and diets) whereas the mucosal virome/phageome are more narrowly but robustly influenced by host pathophysiology (intestinal inflammation).

### Alterations in the diversity and composition of the ileal mucosal virome in CD

In recognition of the significant effect of intestinal inflammation on the ileal virome, we next explored the compositional difference in the ileal virome between CD and HC. We first compared the ileal virome diversity (both α diversity [measured by richness and Shannon index] and β diversity), and found that patients with CD had a remarkably decreased virome richness (a metric of α diversity by calculating the number of viral taxa) compared with HC in both the Guangzhou and Kunming cohorts (*t*-test, $p < 0.05$, Fig. 2a), indicating that the ileal virome in CD has a substantial depletion of viral lineages compared to HC. Meanwhile, principal coordinates analysis (PCoA) was performed to evaluate the β diversity of ileal viromes (inter-individual virome dissimilarities across participants, Fig. 2b). CD viromes showed a significantly different clustering from HC viromes in both the Guangzhou and Kunming cohorts, shifting towards the same direction along the

PCoA portioning (*PERMANOVA* test, both FDR $p < 0.05$, Figs. 2b2 and 3). These data suggest that the composition of the ileal virome at the community structure level is significantly different between CD and HC. In addition, we observed a significant geography effect in shaping the ileal virome composition between the Guangzhou and Kunming cohorts (*PERMANOVA* test, $p < 0.05$, Fig. 2b4), congruent with the geography effect we previously observed in faecal virome[11].

We then assessed the impact of the disease course (CD flare-up versus remission) on the ileal virome (via redundancy analysis [RDA], Fig. 2c). The result showed that while ileal virome in both the CD flare-up and remission groups shifted against HC ileal virome, CD flare-up and remission virome displayed separate shifts in the RDA-PCA plotting (Fig. 2c). Meanwhile, CD flare-up virome were more different from HC virome compared to the difference between CD remission virome and HC virome (Fig. 2c). These data suggest that the ileal virome was compositionally different between CD patients with flare-up versus remission. We therefore stratified CD patients into flare-up and remission groups and then investigated the difference in the ileal virome composition between the two patient groups and the HC group. The richness of the ileal virome was significantly decreased in CD flare-ups than HC in both the Guangzhou and Kunming cohorts, and the Shannon diversity was significantly decreased in CD flare-ups than HC in the Kunming cohort (Mann–Whitney test, all $p < 0.01$, Fig. 2d, e). To further identify the viral taxa associated with CD flare-up and remission respectively, we performed *MaAsLin2* analysis (Microbiome Multivariable Associations with Linear Models 2 with the Benjamini and Hochberg false discovery rate (BH-FDR) approach, $q < 0.2$ [$p < 0.05$]) on the ileal virome, controlling for medications and dietary habit. Echoing the decreased ileal virome richness in CD versus HC, an array of feature viral taxa (mostly bacteriophages) was significantly decreased in both CD flare-up and remission compared to HC; the trend was more pronounced in CD flare-up than remission (at the

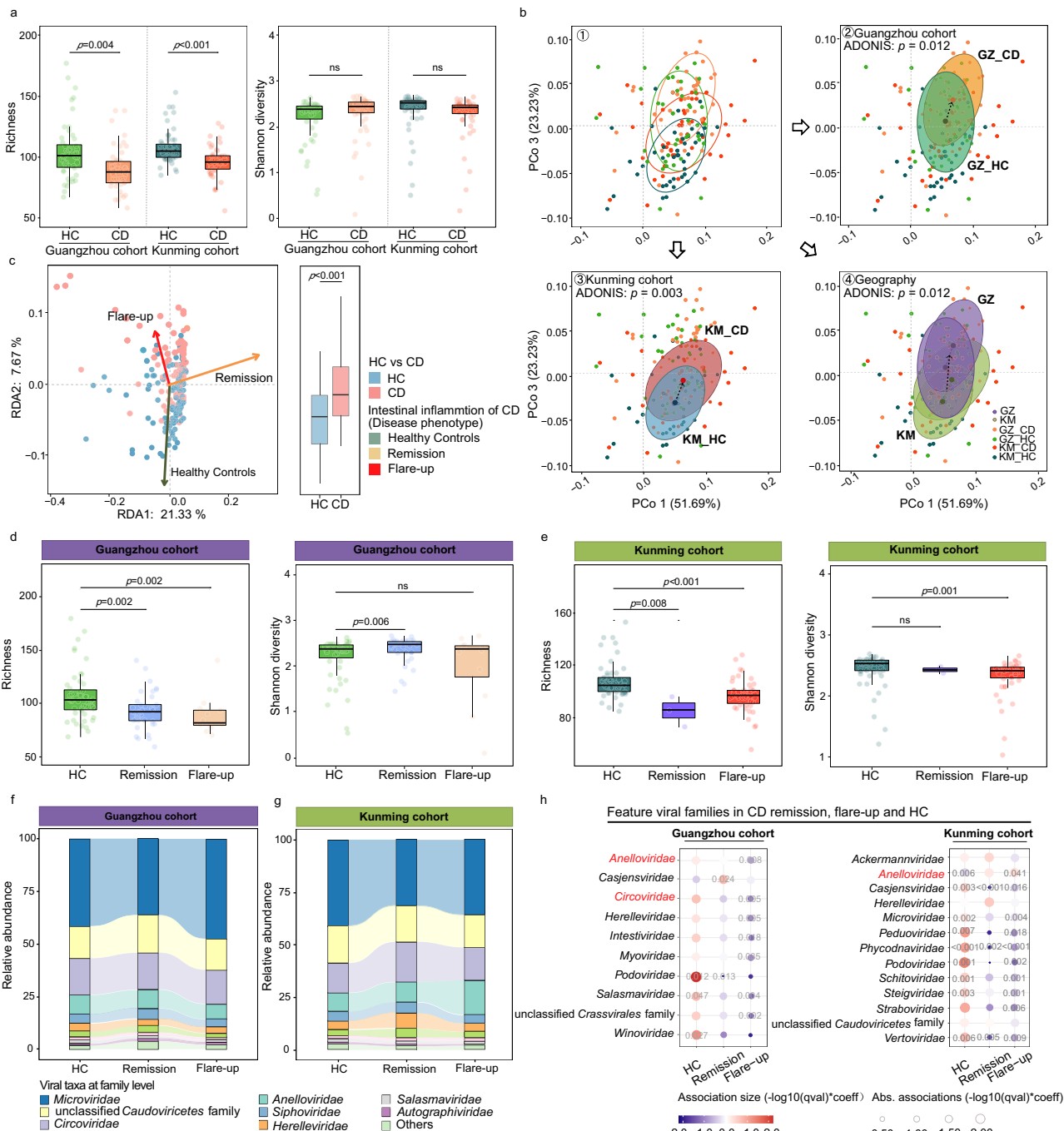

**Fig. 2 | Alterations in the diversity and composition of the ileal virome in CD.**
**a** The α diversity of the ileal virome at the genus level compared between CD and HC in the Guangzhou ($n = 102$) and Kunming cohorts ($n = 106$) respectively. Statistical significance was determined by *t*-test. **b** Beta-diversity visualisation of the ileal virome across HC, CD, and geography. The virome of each subject was analysed and plotted via principal coordinates analysis (PCoA) based on the Bray−Curtis dissimilarity. All statistical significance was determined by *PERMA-NOVA* test. GZ Guangzhou, KM Kunming. **c** Distance-based redundancy analysis (RDA) of the ileal virome variations with intestinal inflammation. Effect size ($R^2$) and direction were visualised via a biplot (depicted in solid arrows). $R^2$ was adjusted with permutation approach, and statistical significance was determined by permutation test. Boxplot shows the comparison of subject distributions on the RDA2 axis between HC and CD. Statistical significance was determined by *t*-test, with 208 independent samples. The α diversity of the ileal virome at the genus level compared among HC, CD remission and flare-up in the Guangzhou (**d**, $n = 102$) and

Kunming (**e**, $n = 106$) cohorts. Between-group comparisons were conducted by Mann−Whitney test. Sankey plot of the relative abundance of the top 10 viral families, compared among HC, CD remission and flare-up in the Guangzhou (**f**) and Kunming (**g**) cohorts. **h** The ileal viral families featured in CD remission and flare-up respectively compared to HC, identified by *MaAsLin2* controlling for medications and dietary habit. The enriched viral families are plotted in red and the decreased viral families are shown in blue. The bubble size and bubble shading indicate the magnitude of the correlation between the viral species and the intestinal inflammation. Viral taxa colour-coated in black denote prokaryotic viruses/bacteriophages, whilst those colour-coated in red denote eukaryotic viruses. Statistical significance was determined by a two-tailed multivariable association test, adjusted by the Benjamini and Hochberg false discovery rate (BH-FDR) adjustment. BH-FDR $q < 0.2$ ($p < 0.05$) was considered significant. For box plots, the boxes extend from the 1st to the 3rd quartile (25th to 75th percentiles), with the median depicted by a horizontal line.

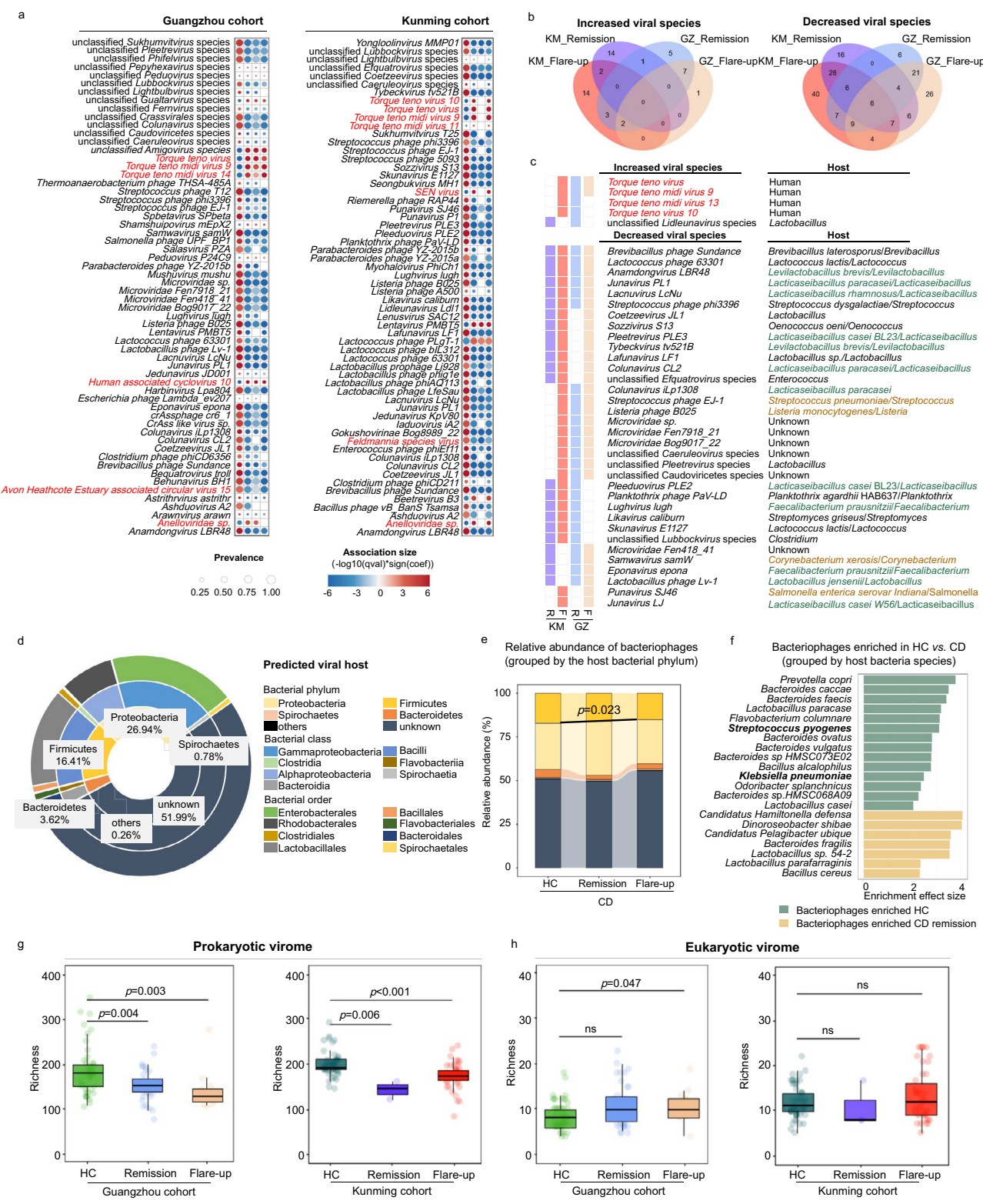

family level: Fig. 2f–h; at the genus level: Supplementary Fig. 4). However, the CD patients in the Guangzhou and Kunming cohorts had two distinct sets of depleted viral taxa respectively (at the family level: Fig. 2h; at the genus level: Supplementary Fig. 4), suggesting that there was a prominent cohort (geography)-specific effect in the configuration of the ileal virome in CD. Of the feature viral families, *Podoviridae* was the only family that was shared between the Guangzhou and Kunming cohorts consistently showing a decreased abundance in both

the CD flare-up and remission groups compared to the HC group (all *p* < 0.05, Fig. 2h). By contrast, *Gammatorquevirus* (a eukaryotic viral genus from the *Anelloviridae* family) was increased in both the CD flare-up and remission groups compared to the HC group, across the Guangzhou and Kunming cohorts (all *p* < 0.05, Supplementary Fig. 4). As *Gammatorquevirus* was found to be elevated in immunocompromised people[15], the increase of *Gammatorquevirus* at the ileal mucosa of CD patients relative to HC highlights that the dysregulation of the

**Fig. 3 | Depletion of prokaryotic viruses and expansion of eukaryotic viruses in CD. a** Altered ileal viral species in CD (remission and flare-up) compared to HC in the Guangzhou (left) and Kunming (right) cohorts, identified via *MaAsLin2* analysis controlling for medications and dietary habit. Only the top 60 significant associations (BH-FDR $q < 0.2$ [$p < 0.05$]) were plotted. The enriched viral species are plotted in red and the decreased viral families are shown in blue. The bubble size denotes the prevalence rate of the viral species across study participants, and bubble shading indicates magnitude of the correlation. Viral taxa colour-coated in black denote prokaryotic viruses/bacteriophages, whilst those colour-coated in red denote eukaryotic viruses. Statistical significance was determined by a two-tailed multivariable association test, adjusted by the BH-FDR adjustment. **b** Venn diagram of the number of shared viral species between groups. GZ Guangzhou, KM Kunming. **c** Feature viral species (increased or decreased in relative abundance) in CD compared to HC, plotted with respect to CD remission, flare-up, and cohorts. The host of each feature species were depicted. For the bacterial hosts, probiotic bacteria were colour-coated in green, and pathogenic bacteria were colour-coated in orange. **d** Taxonomic distribution of the host bacteria for the ileal mucosal bacteriophages, ranked by the relative abundance of their predator bacteriophages. **e** The relative abundance of bacteriophages categorised by the host bacterial phyla in HC, CD remission and flare-up. Between-group comparison was conducted by *t*-test. **f** Differentially enriched bacteriophages, classified according to their host bacteria species, between HC and CD. Enriched species were identified via *LEfSe* analysis (LDA > 2, KW *p* value < 0.05), albeit no specific species were identified in CD flare-up. Host bacteria prediction was based on CRISPR-spacer analysis on viral contigs in *PHYBOX (CHERRY)*[16]. The richness of ileal prokaryotic virome/bacteriophages (**g**) and ileal eukaryotic virome (**h**) in CD remission and flare-up compared to HC in the Guangzhou (left, $n = 102$) and Kunming (right, $n = 102$) cohorts. Between-group comparison was conducted by Mann–Whitney test. For box plots, the boxes extend from the 1st to the 3rd quartile (25th to 75th percentiles), with the median depicted by a horizontal line.

intestinal mucosal immunity in CD might be associated with a bloom of certain eukaryotic viruses. This is also exemplified by our observation that *Cyclovirus* (another eukaryotic viral genus) was markedly expanded in CD flare-up than HC (all $p < 0.001$, Supplementary Fig. 4). Together, these findings indicate that the ileal virome of CD patients had a significant loss of bacteriophage taxa along with an expansion of certain eukaryotic viruses, the pattern of which may get exacerbated over the disease course (flare-up versus remission), especially when considering the intermittent flare-up and remission nature of CD.

To gain a fine-grained view of CD mucosa-associated viruses and pinpoint the viral lineages associated with CD flare-up, we subsequently inspected the ileal virome between CD and HC at the species level. Again, a large number of viral species ($n = 186$; 98.9% were bacteriophages) were depleted in CD patients (in both flare-up and remission) compared to HC in both the Guangzhou and Kunming cohorts (Fig. 3a, b and Supplementary Fig. 5e). Albeit, there were a number of increased viral species in CD flare-up and remission (mostly were unique to each cohort: 10 species in Guangzhou flare-up, 18 species in Guangzhou remission, 21 species in Kunming flare-up, and 17 species in Kunming remission; as enumerated in Fig. 3b). Interestingly, of the increased viral species in CD versus HC, a panel of *Torque teno virus* lineages (−1, −9, −13, and −10) belonging to the eukaryotic *Anelloviridae* family were significantly enriched in CD ileal mucosa, particularly in flare-up mucosa (Fig. 3c and Supplementary Fig. 6a, b). The discordant presence of bacteriophages and eukaryotic viruses in CD versus HC prompted us to investigate the ecological difference in mucosal virome between CD and HC. A significant decrease in the α diversity of the ileal prokaryotic virome was observed in CD versus HC, particularly in CD flare-up versus HC (richness decreased in the Guangdong cohort, $p < 0.01$; both the richness and Shannon diversity decreased in the Kunming cohort, all $p < 0.05$ respectively; Fig. 3g and Supplementary Fig. 5a, b). In contrast, a significant increase in the α diversity of the ileal eukaryotic virome was observed in CD flare-ups versus HC (both the richness and Shannon diversity increased in the Guangdong cohort, all $p < 0.05$; the Shannon diversity increased in the Kunming cohort, $p < 0.05$; Fig. 3h and Supplementary Fig. 5c, d). We also investigated the impact of CD on the ileal bacteriome and found that the CD mucosal bacteriome was markedly different from that of HC, where a significant expansion of Proteobacteria and a reduction of Firmicutes were observed in both Guangzhou and Kunming cohorts (Supplementary Fig. 7a, b, d). Additionally, CD patients in the Guangzhou cohort exhibited an additional depletion of Bacteroidota and Desulfobacterota, compared to the Kunming cohort (Supplementary Fig. 7a, b, d), suggesting a geographic effect in shaping the CD mucosal bacteriome.

To understand the parasitic nature of the bacteriophages that were depleted in CD ileal mucosa in association with the disease course, we searched the host of these CD-depleted bacteriophages against the Virus-Host DataBase. A majority (12/34; 35.29%) of CD-depleted bacteriophage species shared between the Guangzhou and Kunming cohorts were those infecting probiotic bacteria from the Firmicutes phylum, such as *Junavirus PL1*, *Lactobacillus phage Lv-1*, *Tybeckvirus tv521B*, and *Lughvirus lugh* (Fig. 3c, Supplementary Data 2 and Supplementary Fig. 7e). Meanwhile, we also performed viral host prediction based on CRISPR spacers analysis by *PHYBOX (CHERRY)*[16] to investigate the host bacteria of ileal mucosal bacteriophages. This analysis revealed that the predominant hosts of ileal bacteriophages were Proteobacteria, Firmicutes, and Bacteroidetes (ranked in decreasing order according to the abundance of their predator bacteriophages, Fig. 3d), and the bacteriophages infecting Firmicutes were significantly decreased in relative abundance in CD flare-up compared to HC (*t*-test, $p = 0.023$, Fig. 3e). Our further correlation analysis on these CD-depleted bacteriophage species and Firmicutes showed a ubiquitous inverse bacteriophage-bacteria relationship across the HC, CD remission and flare-up groups, yet differing in correlation robustness and effect size (Supplementary Fig. 7c). These data suggest that though a prey-predator relationship might exist between bacteriophages and bacteria at the intestinal mucosa, the inverse relationship is non-synchronised per se which may underpin the disease course of CD.

Meanwhile, we also noted several CD-depleted bacteriophage species whose hosts were pathogenic bacteria, including *Jedunavirus KpV80* (host: *Klebsiella sp.*), *Punavirus SJ46* (host: *Salmonella enterica serovar Indiana*), *Listeria phage BO25* (host: *Listeria monocytogenes*), *Samwavirus samW* (host: *Corynebacterium xerosis*), and *Streptococcus phage EJ-1* (host: *Streptococcus pneumoniae*) (Fig. 3c and Supplementary Fig. 7e). Consistently, based on CRISPR-spacer host prediction by *PHYBOX (CHERRY)*[16], we found that viruses infecting specific pathogens (*Klebsiella pneumoniae* and *Streptococcus pyogenes*), were significantly decreased in CD versus HC (*LEfSe* analysis, KW *p* value < 0.05, Fig. 3f). Many of these host pathogens have been associated with intestinal inflammation and/or IBD exacerbation, such as *Klebsiella*, *Salmonella*, and *Listeria monocytogenes*[17–19]. Although we did not find significant correlations between these bacteriophages and the concerned bacterial pathogens, the depletion of bacteriophages against the pathogenic bacteria at the CD mucosa, particularly in flare-up patients (Supplementary Fig. 7e), may lead to an immune void against pathogens and hence an exacerbated disease progression.

## Functionality alterations of the ileal mucosal virome in CD

We next sought to understand the functionality profile beyond the compositionality of the ileal virome in both health and CD. By *MaAsLin2* analysis, we identified the ileal virome function modules that were differentially enriched in HC, CD flare-up and remission. In HC, the ileal virome was characterised by a vastly diverse presence of viral function categories, including virus-inherent functions (viral replication, package, lysogenic/lysis decision, mobilisation, integration, adhesion and invasion) and host-dependent functions (metabolism,

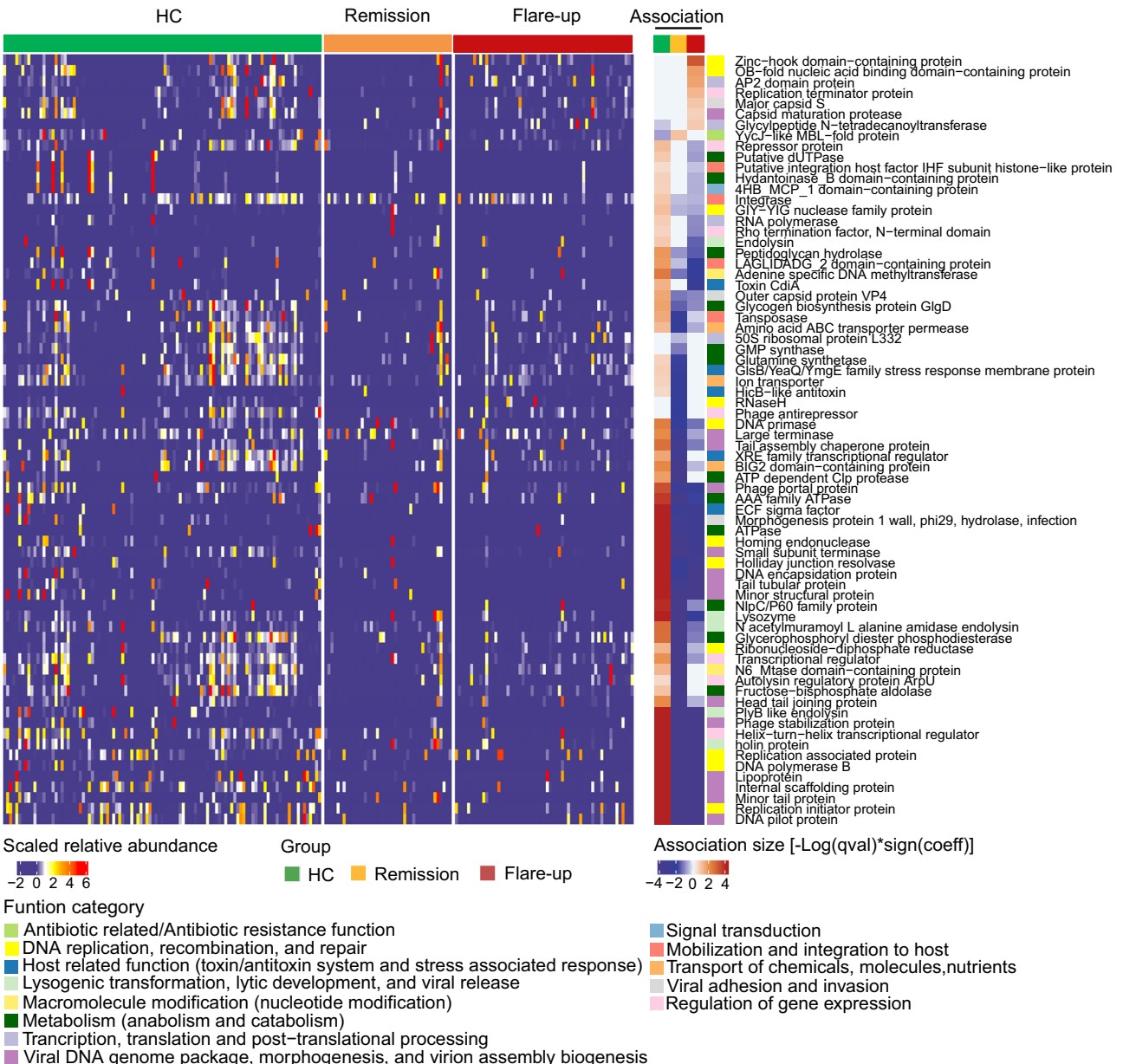

**Fig. 4 | Functionality alterations in the ileal mucosal virome in CD.** The differential virome functions between CD and HC were identified via *MaAsLin2* controlling for medications, dietary habit, and geography. Only function modules with BH-FDR $q < 0.2$ ($p < 0.05$) were shown. Statistical significance was determined by a two-tailed multivariable association test, and adjusted by the BH-FDR adjustment.

signal transduction, macromolecule modification, antibiotics-related/resistant functions, toxin/antitoxin system, stress-associated response, and regulation of gene expression, translation and post-translation modification) (Fig. 4). However, the functionality diversity was significantly abated in the ileal virome of CD versus HC, demonstrated by the under-representation of a large number of viral function modules in both the CD remission and flare-up groups compared to the HC group (Fig. 4). Of note, the viral functions involved in virion assembly biogenesis, lysogenic-to-lytic transformation, and viral release processes (mostly of bacteriophage origin) were significantly down-regulated in CD versus HC virome (Fig. 4), suggesting a compromised viral reproduction in CD. Consistently, we found a significant decrease in the viral richness of both lytic and temperate bacteriophages in the ileal virome of CD versus HC, evident in both the Guangzhou and Kunming cohorts (*t*-test, all $p < 0.01$, Supplementary Fig. 7f). In addition, the viral functions related to viral adhesion/invasion, DNA mobilisation/integration, and metabolism were also significantly abolished in the ileal virome of CD versus HC (Fig. 4). A large proportion of the depletions of these concerned viral functions were associated with both the bacteriophages depleted in CD ileal virome (species colour-coated in blue in Supplementary Fig. 8) and bacteriophages which were not differentially abundant between CD and HC (as demonstrated by the associations between bacteriophages species, colour-coated in black, and depleted viral functions in CD, shown in Supplementary Fig. 8). These data together imply that the functionality changes in CD ileal virome are intertwined with the changes in both the species abundance of viral taxa and potentially the genetic makeup of individual bacteriophages.

By contrast, 7 viral functions were elevated in relative abundance in the ileal virome of CD flare-up versus HC, including Zinc-hook domain-containing protein and OB-fold nucleic acid binding domain-containing protein (both are DNA replication, recombination, and repair-associated functions), AP2 domain protein and Replication terminator protein (both are protein translation and expression-

associated functions), major Capsid S (eukaryotic virus structural protein), Capsid maturation protease (viral capsid maturation-associated process), and Glycylpeptide N-tetradecanoyltransferase (N-myristoylation of cellular and viral proteins) (Fig. 4). This data indicates that the viral functions enriched in CD ileal virome may enhance the generic function of the viruses and/or their hosts. Interestingly, the elevation of the Glycylpeptide N−tetradecanoyltransferase-coding gene in the CD virome was correlated with the presence of *Torque teno viruses* (Supplementary Fig. 8). N-myristoylation of ADP-ribosylation factor 6 (ARF6) was reported to maintain ARF6 on cell membranes during the GTPase cycle, and ARF6 played critical roles in cellular endocytosis, epithelial barrier disruption, and intestinal inflammation[20,21], all of which are important functions in host defence against eukaryotic viral infection. The correlation between Glycylpeptide N-tetradecanoyltransferase and *Torque teno viruses* might indicate an altered host cell biology in relation to the aberrant viral presence at the ileal mucosa of CD patients. Overall, the metabolism and phenotype of both the prokaryotic hosts (bacteria) and the eukaryotic hosts (namely human cells herein) are hypothesised to be altered in CD versus HC, considering that frequent infection and integration of bacteriophages/viruses and their functional genes into bacteria and human cells are anticipated in the gut[22,23].

## Trans-kingdom interaction between ileal virome and bacteriome in CD

Given the intimate relationship between viruses (bacteriophages and eukaryotic viruses) and their hosts (bacteria and eukaryotic cells) and the intermediating effect of host immunity between trans-kingdom microbes[3,24,25], the gut virome-bacteriome ecological network were posited to be altered at the intestinal mucosa in patients with CD compared to HC. We hence characterised the trans-kingdom interactions between the ileal virome and bacteriome in CD (remission and flare-up) versus HC. At the microbiome community level, we assessed the correlation between the α diversity metrics of the ileal virome and bacteriome. We found robust positive intra-kingdom correlations within the α diversity metrics of the virome and bacteriome respectively, which were consistently present across the HC, CD remission and flare-up groups (all $p < 0.01$, Spearman correlation $R > 0.15$, Fig. 5a). In addition, significant positive trans-kingdom correlations between the richness of virome and the Shannon, Simpson, and evenness index of bacteriome were observed in HC (all $p < 0.05$, Fig. 5a). However, such a correlation pattern was lost in both CD remission and flare-up (Fig. 5a). Similarly, significant inverse trans-kingdom correlations between the richness of bacteriome and the Shannon, Simpson, and evenness index of virome were observed in HC and CD remission, the correlation pattern of which was also lost in CD flare-up (all $p < 0.05$, Fig. 5a). These data suggest that the virome richness and the bacteriome richness are both the bedrock for a homoeostatic mucosal microbiome ecology, and the loss of them might be associated with intestinal pathophysiology and CD flare-up.

To obtain a landscape view of the sophisticated interactions between the mucosal virome and bacteriome, we further evaluated the trans-kingdom correlations between the top abundant 50 bacteriophage species of the ileal virome and the top abundant 30 bacterial genera of the ileal bacteriome (Fig. 5b). In our analysis, a large number of bacteriophages-bacteria correlations were seen in HC (Fig. 5b), suggesting an intense and extensive cross-kingdom interaction between mucosal phageome and bacteriome at steady-state. However, the correlation network became weakened in both CD remission and flare-up, as demonstrated by a remarkably decreased number of bacteriophage-bacteria correlations as well as decreases in the correlation effect sizes, compared to the correlation network observed in HC (Fig. 5b). For instance, correlations between ileal bacteriophages and the bacterial genera *Bifidobacterium*, unclassified Lachnospiraceae, *Cellulosimicrobium*, *Bacteroides*, and *Streptococcus* seen in HC

were depleted or attenuated in CD (Fig. 5b). By contrast, inverse correlations between bacteriophages and *Prevotella* were enriched in CD (particularly in CD with flare-ups) versus HC, of which multiple *Prevotella* species were reported to have pro-inflammatory effect in intestinal inflammation[26]. The weakened phageome-bacteriome network in CD versus HC was attributable to both CD-depleted ($n = 23$) and CD-non-depleted bacteriophage ($n = 27$) species (Fig. 5b). These data indicate that the weakened phageome-bacteriome ecological network in the ileal mucosa of CD might be ascribed to the loss of bacteriophage species in CD and beyond that, to the altered bacteriophage-bacteria interactions mediated by non-depleted species in CD (presumably through phenotypic changes in bacteria).

Surprisingly, the weakened phageome-bacteriome network observed in CD was more pronounced in CD remission compared to CD flare-up, supported by the observation that the phageome-bacteriome network in the CD remission group had a more decreased number of significant correlations as well as decreased correlation effect sizes (Fig. 5b). This finding also applied to the eukaryotic virome-bacteriome interactions (Supplementary Fig. 9a). These data indicate that the mucosal viruses/bacteriophages and bacteria may not establish an equilibrious ecological network in lock-step with each other during the remission phase after a fulminant active disease course (flare-up) in CD. Furthermore, macroscopic (endoscopic) remission of CD does not virtually reflect microscopic (histologic) remission, where an underlying quiescent abnormality in host physiology at the cellular and/or molecular level may be fuelling another flare-up episode of intestinal inflammation[27]. Therefore, ecological remission at the virome-bacteriome level should also be considered one of the clinical goals in CD therapeutics.

## The impact of medications on ileal virome composition

Medications are a critical factor in influencing the faecal microbiome composition[28], and serve as a significant confounder in gut microbiome interrogation studies in a disease of concern. However, the impact of medications on the gut virome at the mucosal level was unknown. Our metadata-virome analysis discovered that medications overall explained 1.1% of the ileal virome variance (VPA analysis, *permutation* test, $p < 0.05$, Supplementary Fig. 4a), indicating a modest yet significant impact on the ileal virome composition. We then separately interrogated the impact of 4 commonly prescribed CD-related medications (biologics, immunosuppressants, glucocorticoids, and 5-ASA) on the ileal virome composition. Among 103 CD patients, 77 (74.76%) were treated with one or more first-line/second-line medications in the preceding 3 months prior to the date of sampling, including biologics ($n = 40$, 38.83%), immunosuppressants ($n = 33$, 32.03%), glucocorticoids ($n = 22$, 21.36%) and 5-ASA ($n = 49$, 47.57%) (Table 1 and Supplementary Data 1). Through *MaAsLin2* analysis, we discerned the viral species that were associated with each medication, while controlling for geography and diet. A total of 34 meditation-virus associations were discovered via the analysis, where 28 viral species were identified as significantly associated with these medications (Fig. 6a). Amongst the significant medication-virus associations ($n = 34$), 15 (44.1%) were ascribed to biologics, 13 (38.2%) were ascribed to glucocorticoids, 4 (11.8%) were ascribed to immunosuppressants, and 2 (6.2%) were ascribed to 5-ASA, suggesting that biologics had the largest effect size in influencing the ileal virome composition. Amongst the medications-associated viral species ($n = 28$), 14 (50%) were specific to medications, whereas the remaining 14 (50%) were associated with both medications and CD (Fig. 6a). Intriguingly, of the viral species associated with both medications and CD, 7 species were oppositely associated with biologics and CD, 3 species were oppositely associated with glucocorticoids and CD, 1 species was oppositely associated with immunosuppressants and CD, and 1 species was oppositely associated with 5-ASA and CD (Fig. 6a). These data indicate that medications may have a rectification effect against CD-distorted virome, and biologics

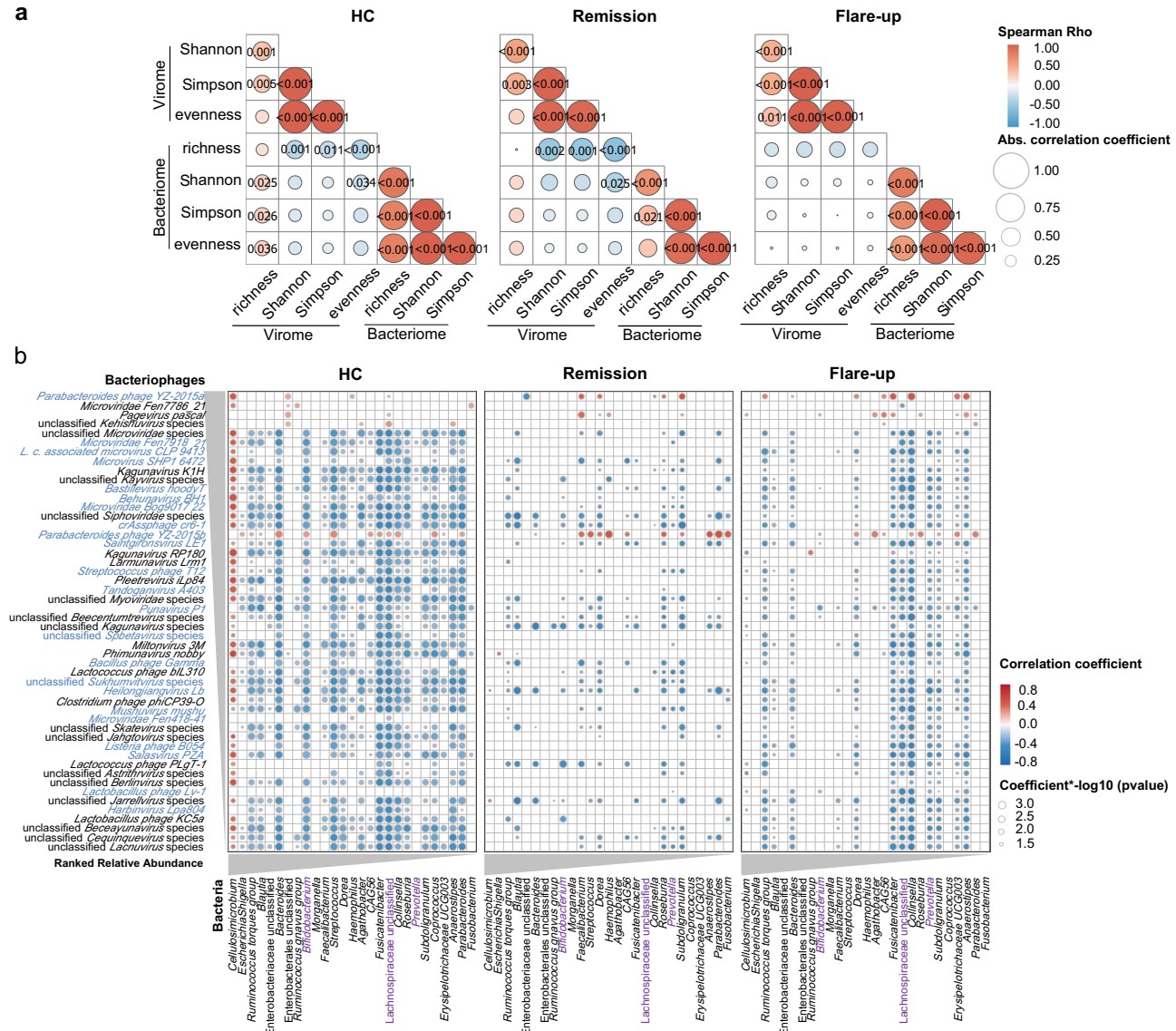

**Fig. 5 | Trans-kingdom interactions between ileal viruses and bacteria.**
**a** Correlations between the α diversity metrics (richness, diversity [Shannon index and Simpson index], and evenness) of the ileal virome and bacteriome in HC, CD remission and flare-up, respectively. Spearman's correlation coefficient was calculated, and statistical significance was determined for all pairwise comparisons.
**b** Trans-kingdom correlations between the top 50 bacteriophages at the species level and the top 30 bacterial genera in HC, CD remission and flare-up, respectively. The correlations were calculated with *FastSpar*, and statistical significance was

determined for all pairwise comparisons by a bootstrap method. Only statistically significant correlations were plotted. Red bubbles indicate positive correlations and blue bubbles indicate inverse correlations. The size and shading intensity are proportional to the magnitude of the correlation. Bacteriophages colour-coded in blue denote the species depleted in CD versus HC. Bacterial genera colour-coded in purple denote those genera also appeared in the bacteriome-virome ecology analysis in our later murine study (shown in Fig. 7j) to establish a causal effect and validate the finding in humans.

may have the largest effect. Consequently, biologics overall restored the CD virome-bacteriome network to a more similar level to that in HC, compared to immunosuppressants, glucocorticoids and 5-ASA (Supplementary Fig. 9b).

To further explore the causal versus consequence relationship between CD medications and gut bacteriophages, we harnessed an in vitro model where we incubated faecal microbiota preparations from 5 healthy individuals with medications of interest (5-ASA, immunosuppressant drug azathioprine [AZA], and glucocorticoid drug methylprednisolone [MP], respectively), and then specifically probed the post-incubation abundance changes in those bacteriophages that were observed to be associated with CD medications in human patients. While a majority of bacteriophages were not influenced by the 3 medications of interest, we found that a number of bacteriophages were impacted by 5-ASA and MP (*Behunavirus BH1* was increased by 5-

ASA, Supplementary Fig. 10b; *Brevibacillus phage Sundance* and *Lactobacillus phage Lj928* were decreased by MP, Supplementary Fig. 10c, f), exhibiting an abundance change in the same directions seen in the medication-virome associations in patients. This data suggest that CD medications can causally change the abundances of a defined set of bacteriophages, yet a lot of bacteriophage species may not causally vary as a function of medication use. To further substantiate the impact of these medications on intestinal bacteriophages under physiological conditions, we conducted in vivo experiments via administering 5-ASA, AZA, and MP, and then investigated the abundance change in the target gut bacteriophage post medication administration (Supplementary Fig. 11). Again, while the majority of the bacteriophages remained unchanged by these medications, a subset of bacteriophages were impacted by 5-ASA and MP, coinciding with the findings in our in vitro experiment (decreased *Lactobacillus phage*

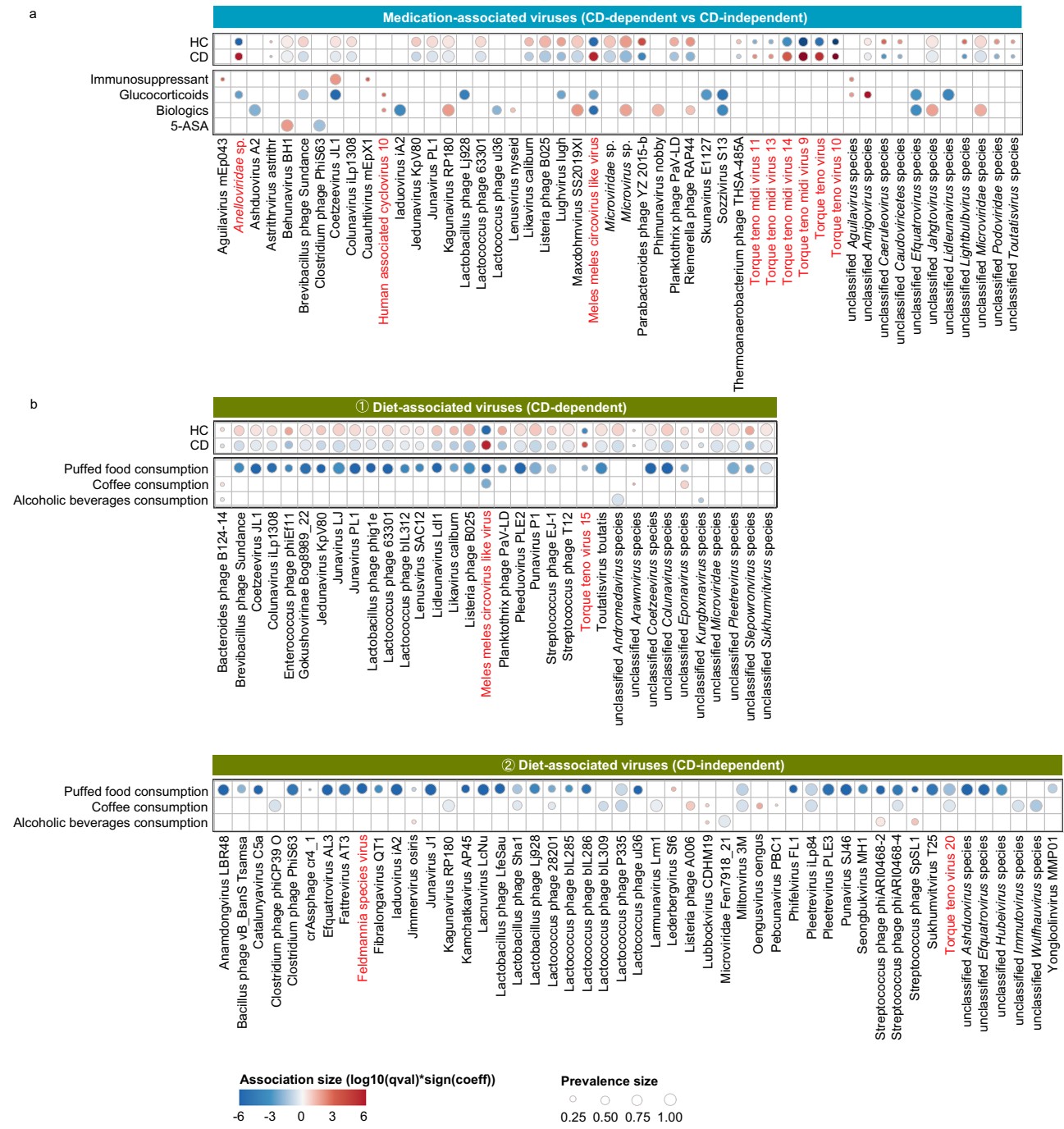

**Fig. 6 | Medications- and diet-associated ileal viral species. a** Medications-associated ileal viral species. Associations between viral species and medications were identified via *MaAsLin2* controlling for remission/flare-up, dietary habit, and geography. Only the significant associations (BH-FDR $q < 0.2$ [$p < 0.05$]) were plotted via bubble plot. The red bubbles indicate species with a positive association with medications, and blue bubbles indicate species with an inverse association with medications. The bubble shading intensity indicates the magnitude of the association between viral species and medications. The bubble size indicates the prevalence rate of the viral species across study participants. The associations of these viral species with CD were also plotted for comparative visualisation. Statistical significance was determined by a two-tailed multivariable association test, adjusted by the BH-FDR adjustment. **b** Diet-associated ileal viral species.

*Lj928* with MP use) and in our human observations (decreased *Clostridium phage PhiS63* with 5-ASA use, and decreased *Coetzeevirus JL1 phage* with MP use) (Supplementary Fig. 11b, f, h). Overall, these data together substantiated that medication use has a critical impact on shaping the gut phageome.

We also evaluated the effect of biologics, immunosuppressants, glucocorticoids, and 5-ASA on the functionality profile of the ileal virome. The majority of viral functions that were inversely associated with these medications were involved in biological processes related

to DNA replication, recombination, and repair, as well as to viral DNA package, morphogenesis, and virion assembly (Supplementary Fig. 12a). Notably, biologics, glucocorticoids, and 5-ASA showed inverse correlations with the concerned viral functions whereas immunosuppressants showed positive correlations (Supplementary Fig. 12a). These data suggest that most of the existing medications in CD therapeutics may potently inhibit virus replication and assembly. Though glucocorticoids and immunosuppressants all have immunosuppressing roles in CD therapeutics, they differed in association

directions with the functions of virus replication and assembly (Supplementary Fig. 12a). Mechanistically, while glucocorticoids target a broad spectrum of leucocytes, immunosuppressants target primarily the T-cell activation cascade[29]. It indicates that medications with analogous roles may have disparate effects on the functionality profile of the mucosal virome, likely dependent on the various immune-tuning mechanisms of each medication. In contrast to biologics which showed the largest effect size on the taxonomic composition of the ileal virome (Fig. 6a), glucocorticoids had the largest effect size on the functionality profile of the ileal virome (Supplementary Fig. 12a). Such disconcordant effect sizes of medications on the compositionality and functionality of the ileal virome imply that different medications have discrepant roles in modulating the taxonomic versus functional diversity of the mucosal virome.

## Diet-associated ileal virome features

Diet is an important factor in shaping the gut microbiome and was reported to contribute 20% of the faecal microbiome variations in humans[30], while the impact of diet on intestinal mucosal virome is largely unclear. In our metadata-virome correlation analysis in Fig. 1g, out of 23 dietary components, the consumption of alcoholic beverages, coffee, and puffed food respectively showed a significant association with the ileal virome variations. Thus, we next explored individually the association of alcoholic beverages, coffee, and puffed food with the composition and function of the ileal virome. First, we evaluated the relationship between the consumption frequency of the dietary component of concern and the α diversity of the ileal virome, or between consumers/non-consumers of the dietary component of concern and the α diversity of the ileal virome. The results showed that a weekly intake of coffee had an increased richness of the ileal virome when compared to those individuals who had a lower coffee-intake frequency ($p < 0.05$, Supplementary Fig. 13c, d). In contrast, consumers of puffed food showed a significant decrease in both the richness and diversity of the ileal virome than non-consumers of puffed food (both $p < 0.01$, Supplementary Fig. 13e, f). These data suggest that coffee drinking and puffed food consumption have different impacts in the compositional landscape of the ileal virome.

We then identified the viral species associated with alcoholic beverages, coffee, and puffed food respectively, adjusting for disease (CD versus HC), geography and medications. A total of 83 viral species were associated with the consumption of alcoholic beverages, coffee, and puffed food, a majority (79/83, 95.2%) of which were bacteriophages (Fig. 6b); amongst these viral species, 49 viral species (49/83, 59.0%) were dietary component-specific (Fig. 6b1), whilst 34 viral species (34/83, 41.0%) were also concurrently impacted by CD (Fig. 6b2). Overall, puffed food was associated with a substantial depletion of a variety of viral species (Fig. 6b), and a lot of them were also CD-depleted species (Fig. 6b1). Congruently, diet-virome function correlation analysis also showed that puffed food was associated with a substantial depletion of various viral functions, a lot of which were also CD-depleted viral functions related to viral lifecycle and metabolism (Supplementary Fig. 12b). Puffed food is enriched for fat and food additives (including leavening agents, emulsifiers, and preservatives), all of which have been reported to have a detrimental effect on the gut bacteriome and to exacerbate intestinal inflammation[31]. Building on this observation, the detrimental effect of puffed food may be generalised to the ileal virome at the intestinal mucosa level linking to gut inflammation.

By contrast, coffee drinking was associated with a depletion of *Circoviridae sp.* (eukaryotic virus) and an increase of *Bacteroides phage B124-14* and an unclassified *Eponavirus* species (both are bacteriophages) (Fig. 6b2). Interestingly, CD showed the opposite association directions with coffee drinking on the all of these viral species (Fig. 6b2). At the virome functionality level, coffee drinking also showed a significant association with a set of viral functions, again in

the opposite association directions with CD on them (Supplementary Fig. 12b). Prior studies in humans and mice have reported that coffee drinking was a protective factor for IBD[32,33]. Together, our data suggest that the protective effect of coffee drinking against IBD might be partly attributed to its effect on the intestinal mucosal virome on both the composition and function levels. Alcoholic beverage consumption was inversely associated with *Bacteroides phage B124-14*, an unclassified *Andromedavirus* species, and an unclassified *Kungbxnavirus* species (all are bacteriophages, Fig. 6b2). Consistently, CD also showed inverse correlations with these 3 species (Fig. 6b2). The bacteria host of *Bacteroides phage B124-14*, enterotoxigenic *Bacteroides fragilis*, was reported to be increased in CD and to aggravate intestinal inflammation through its protease activity[34,35]. Considering that alcohol is primarily absorbed in the GI mucosa, it may have a deleterious effect on *Bacteroides phage B124-14* linking to a bloom of *B. fragilis* and mucosal inflammation in CD, whilst the cause versus consequence effect remains to be established.

## Human CD ileal VLPs causally exacerbated intestinal inflammation in mice

To pinpoint the cause or consequence relationship between the disturbed small bowel mucosal virome and intestinal inflammation in IBD, we isolated ileal VLPs from patients with CD ($n = 4$) and non-CD controls ($n = 4$) (Supplementary Fig. 14a), and then respectively administered them to mice followed by dextran sulfate sodium (DSS) treatment to elicit intestinal inflammation (a murine IBD model, Fig. 7a). Mice administered CD ileal VLPs exhibited a significantly more severe intestinal inflammation than those administered non-CD ileal VLPs after DSS treatment, as demonstrated by shortened colon lengths, higher histological inflammation scores, and aggravated endoscopic inflammation (Mann–Whitney test, $p = 0.015$, $p = 0.027$, respectively, Fig. 7b–d and Supplementary Fig. 14b, c). Meanwhile, increased mRNA expression levels of pro-inflammatory cytokines and chemokines were also seen in mice administered CD ileal VLPs than in mice administered non-CD ileal VLPs, including *Il-1α*, *Il-1β*, *Ccl3*, and *Ccl8* (Mann–Whitney test, all $p < 0.05$, Supplementary Fig. 14d). These phenotypes were also reproduced in another batch of DSS mice (Supplementary Fig. 15) as well as in a new IBD murine model, TNBS-induced intestinal inflammation (a model more reflective of the CD inflammation, Supplementary Fig. 16). Analogously, in the TNBS model, CD ileal virome administration evoked more pronounced intestinal inflammation in mice, compared to non-CD ileal virome administration (Supplementary Fig. 16). These data together substantiated that ileal VLPs derived from patients with CD exacerbated intestinal inflammation in mice, establishing a causal relationship between the disturbed mucosal virome and aggravation of intestinal inflammation.

To further investigate the signalling pathways that were evoked by CD mucosal VLPs in mice, underlying the exacerbated intestinal inflammation phenotype, we performed RNA-seq on the intestinal tissues from CD-VLPs-administered versus non-CD-VLPs-administered mice. The top up-regulated signalling pathways by CD mucosal VLPs were those involved in defence responses to microbes (viruses/bacteria, exemplified as follows) and host inflammatory responses (including genes coding for *Il1f9*, *Il22*, *Ccl3*, *Ccl4*, *Cxcl3*, *Nfam1* and *Cebpe*) (Fig. 7e, f). Among the up-regulated microbial defence pathways, genes coding for microbial sensing and effector functions were potently up-regulated, including those bacterial- and/or viral-sensors (such as *Siglec-1*, *Lbp*, *Ifi204*, *Ifi209*; these genes are highly expressed in phagocytic/epithelial cells) and those genes for the downstream microbial-defence effector functions (such as *Oas2*, *Fcgr1*, *Nlrp1a*, *Adamts5*, *HP*) (Fig. 7e, f). These data indicate that the augmentation of the host's microbial-sensing and -defence cascade, provoked by CD mucosal VLPs administration, may contribute substantially to the aggravation of intestinal inflammation in mice.

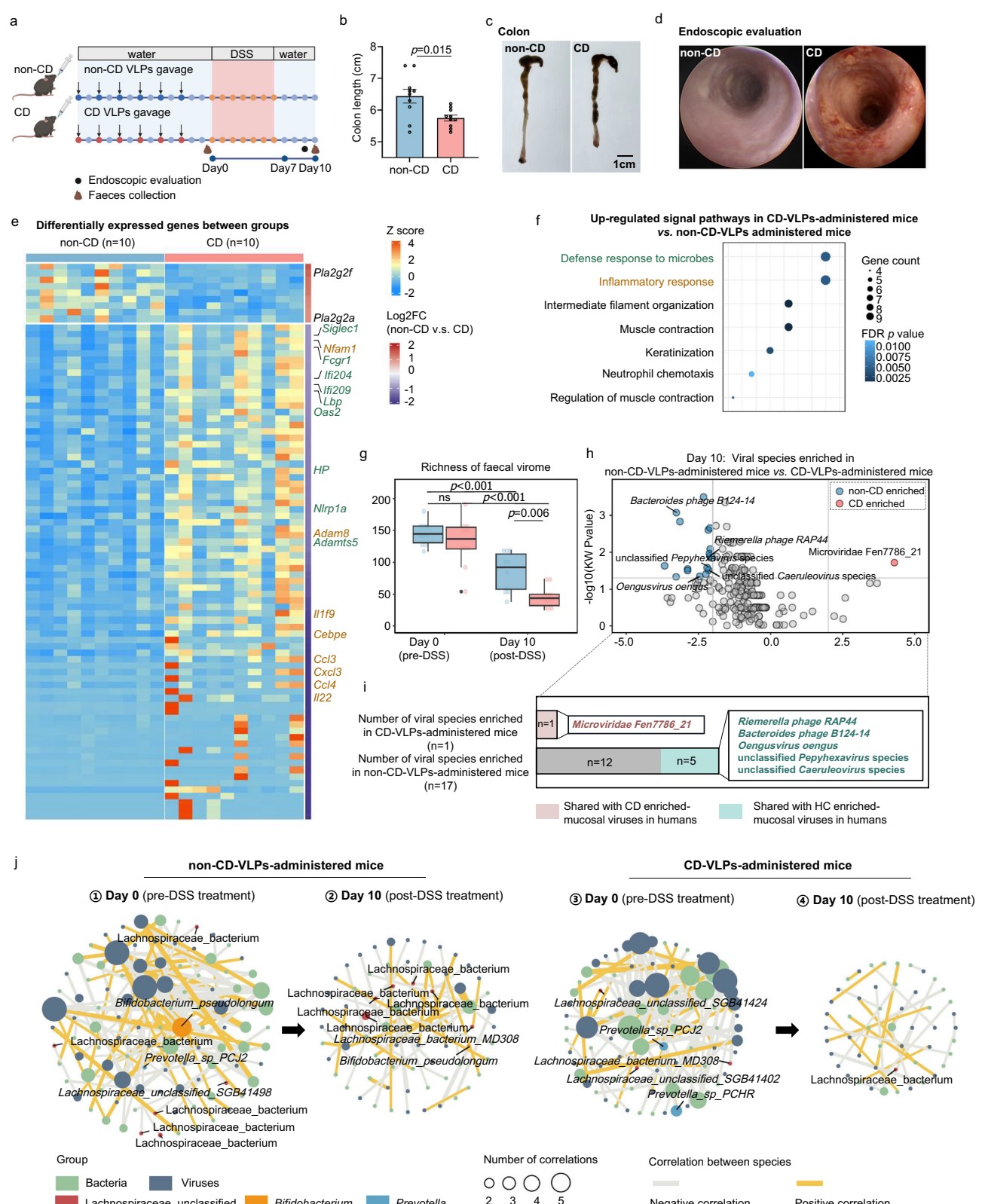

While the significantly up-regulated expressions of microbial sensor genes in CD-VLPs administered mice included those for recognising both bacteria and viruses (such as *Lbp* for recognising bacterial LPS, *Siglec-1* for recognising viral polysaccharides and glycosylated proteins, *Ifi204 and Ifi209* for recognising dsDNA from bacteria and viruses), we then reasoned that the aggravated intestinal inflammation in CD-VLPs-administered mice might be ascribed to the modulation effect of CD-VLPs on the intestinal virome and bacteriome and hence augmented immune responses. In line with this postulation and our findings in human CD patients, CD-VLPs-administered mice displayed a substantially lower richness and diversity of the faecal virome after DSS treatment,

**Fig. 7 | Human CD ileal VLPs exacerbates intestinal inflammation in mice via reshaping the bacteriome-virome ecology and augmenting microbial sensing in the host. a** Schematic of animal experiment (*n* = 10 mice in each group). **b, c** Colon length and representative images from non-CD-VLPs-administered and CD-VLPs-administered mice, represented as mean ± S.E.M. Statistical significance was determined by a two-tailed Mann–Whitney test. **d** Representative endoscopic images from non-CD-VLPs-administered and CD-VLPs-administered mice. **e** Heatmap of differentially expressed genes in the intestines of non-CD-VLPs-administered versus CD-VLPs-administered mice. Genes related to inflammatory response or microbial sensing/defence response were colour-coated in brown or dark green. Statistical significance was determined by a one-tailed Wald test, adjusted by FDR adjustment, and the FDR adjusted *p* < 0.05 were considered statistically significant. **f** Up-regulated signal pathways in CD-VLPs-administered versus non-CD-VLPs-administered mice, analysed via Gene ontology analysis. Statistical significance was determined by a two-tailed Fisher's exact test, adjusted by the FDR adjustment. **g** The richness of faecal virome in non-CD-VLPs-administered and CD-VLPs-administered mice at the pre-DSS and post-DSS timepoints. For box plots, the boxes extend from the 1st to the 3rd quartile (25th to 75th percentiles), with the median depicted by a horizontal line. Between-group comparison was conducted by a two-tailed two-way ANOVA test with Sidak's multiple comparisons test, with ten mice in each group. **h** Volcano plot of differentially enriched viral species between the faeces of non-CD-VLPs-administered and CD-VLPs-administered mice post DSS treatment, identified via *LEfSe* analysis. Statistical significance was determined by a two-tailed Kruskal–Wallis test (LDA > 2, KW *p* value < 0.05). **i** The number of viral species enriched in non-CD-VLPs-administered and CD-VLPs-administered mice at day 10. Those viruses shared with CD patients and health humans were respectively colour-coated in pink and light-green. **j** Trans-kingdom interactions between faecal virome and bacteriome in non-CD-VLPs-administered (**j**1, 2) and CD-VLPs-administered (**j**3, 4) mice. Day 0 virome-bacteriome network (**j**1, 3) represents reshaped trans-kingdom ecology after VLPs gavage, before DSS treatment. Day 10 virome-bacteriome network represents trans-kingdom ecology after intestinal inflammation induction by DSS treatment. The correlations were calculated with R package *ggClusterNet*. Statistical significance was determined for all pairwise comparisons by a bootstrap method, and statistically significant correlations and correlation coefficient >0.15 were plotted.

compared to non-CD-VLPs-administered mice (Mann–Whitney test, *p* < 0.001 and *p* < 0.01, respectively, Fig. 7g and Supplementary Fig. 14e). Moreover, mice with a lower virome diversity had a shorter colon length (a hallmark of intestinal inflammation) (Spearman correlation *R* = 0.5, *p* = 0.03, Supplementary Fig. 14f). Amongst the 17 significantly depleted viruses in CD-VLPs-administered mice, 5 (29.4%) were shared with the underrepresented viruses in CD patients (Fig. 7h, i). Interestingly, both the richness and diversity of the gut virome did not differ between CD-VLPs-administered mice and non-CD-VLPs-administered mice, at day 0 prior to DSS treatment (shown in Fig. 7g and Supplementary Fig. 14e). It prompted us to speculate that administered CD-VLPs may have a crucial effect in re-configuring the gut bacteriome and/or the bacteriome-virome ecology prior to inflammation onset, thereby rendering recipient mice different susceptibilities to DSS-induced intestinal inflammation. Hence, we subsequently explored the changes in the bacteriome composition and bacteriome-virome ecology in mice before (at day 0) and after DSS treatment (at day 10). In favour of our speculation, the faecal bacteriome richness pre-DSS treatment (at day 0) was significantly lower in CD-VLPs-administered mice than in non-CD-VLPs-administered mice (Mann–Whitney test, *p* < 0.001), whilst no significant differences were observed post-DSS treatment (at day 10, Supplementary Fig. 17a, b). Meanwhile, a lower pre-DSS faecal bacteriome richness was associated with a shorter colon length (Spearman correlation *R* = 0.49, *p* = 0.026, shown in Supplementary Fig. 17c, d). These data indicate that CD ileal VLPs reshaped the bacteriome preceding intestinal inflammation upon DSS insult; the extent of bacteriome reshaping correlated with inflammation severity. In addition, our bacteriome-virome interaction network analysis further showed that CD ileal VLPs administration resulted in a weakened bacteriome-virome ecology pre DSS treatment (at day 0) than the network observed in non-CD-VLPs-administered mice (Fig. 7j1). After DSS treatment (at day 10), the impaired bacteriome-virome network was further weakened, more pronounced in CD-VLPs-administered mice than in non-CD-VLPs-administered mice (Fig. 7j2, 4). Overall, in non-CD-VLPs-administered mice, the bacteriome-virome network was characterised by *Bifidobacterium*-virome and Lachnospiraceae-virome correlations, both before (day 0) and after DSS treatment (day 10) (Fig. 7j1, 2). By contrast, these bacteriome-virome interactions predominated by *Bifidobacterium* and Lachnospiraceae were markedly weakened in CD-VLPs-administered mice, as demonstrated by a smaller number of those cross-kingdom correlations. Of note, on day 0 pre DSS treatment, *Prevotella*-virome interactions were strengthened in CD-VLPs-administered mice than non-CD-VLPs-administered mice (Fig. 7j1, 3), further supporting that CD ileal VLPs reshaped the bacteriome-virome ecology. Overall, these data in mice are analogous to our observations in human CD patients that *Bifidobacterium*-virome and Lachnospiraceae-virome interactions were weakened whereas *Prevotella*-virome interactions were strengthened in CD patients compared to healthy controls (Fig. 5b).

## Discussion

CD is an autoimmune disease characterised by intermittent episodes of intestinal mucosal inflammation, whereby gut bacteriome dysbiosis was reported to contribute greatly to the pathogenesis of CD. Apart from bacteriome, a large number of viruses/phages in the gut contribute to defence against pathogen invasions, modulation of the bacteriome ecology and the regulation of the mucosal immunity at intestinal barriers. Disturbance in the mucosal virome is postulated in the present study to underline CD onset and/or disease course. Most of the existing gut virome studies centred on the faecal virome. In those studies concerning faecal viromes in IBD[36,37], both CD and UC patients were shown to have increased virome richness and diversity, and characterised by an expansion of *Caudovirales* yet a reduction in *Microviridae* in abundance, compared to HC. These findings were also observed in the faeces and intestinal tissues of paediatric CD patients based on a modest sample sized cohort[38–40]. In addition, temperate phages were found to predominate the faecal virome in CD compared to HC[41]. However, to date, the mucosal virome, consisting of both phageome and eukaryotic virome, remains unclear in both health and CD, particularly on the small bowel level. Our study for the first time explored the composition and function of the ileal virome (at the small bowel mucosa level) in healthy adults and CD patients, in association with various clinical factors, including medications, diet, and geography. In our study, the ileal virome richness was overall significantly decreased in CD versus HC, and it was more pronounced in patients with flare-ups than remission. Furthermore, the richness of bacteriophages was decreased whereas the richness of eukaryotic viruses was increased, suggesting that bacteriophages and eukaryotic viruses may have discrepant roles in the pathogenesis and the disease course of CD. Importantly, we unveiled the causal effect of CD ileal mucosal virions on intestinal inflammation in IBD mice. The disturbed virome reshaped the gut bacteriome as well as the bacteriome-virome ecology, leading to augmented pro-inflammatory immune responses in microbial sensing/defence and hence aggravation of intestinal inflammation.

Accompanying the increase in richness of the eukaryotic virome, we observed a concordant expansion of *Torque teno viruses* from the *Anelloviridae* family in patients with CD, particularly in those with flare-

ups. In addition, we also found cohort-specific enrichment of different eukaryotic viruses in patients with CD, such as *Human-associated Cyclovirus 10* in the Guangzhou cohort, and *Avon Heathcote Estuary-associated circular virus 15* in the Kunming cohort. Analogously, at the faecal virome level, two separate studies conducted in another Chinese cohort of CD patients ($n = 14$) and a Belgium cohort of UC patients ($n = 64$) also found an increase in eukaryotic viruses in patients compared to healthy controls[37,42]. Though there might be population/geography-specific configurations in the gut virome, as shown in our and others' studies[43–45], the consistently observed bloom in the eukaryotic viruses in IBD patients implies that IBD might be an immune-dysregulated disease associated with chronic viral infections. The augmented richness of the *Anelloviridae* family at the ileal mucosa of CD patients versus HC (particularly in flare-up patients; and independent of immunosuppressants and glucocorticoids medications, Supplementary Fig. 6c, d) provides an appealing link between eukaryotic viruses and active mucosal inflammation in IBD. A recent study found that the protein of *Orthohepadnavirus* (another eukaryotic viruses belonging to the family *Hepadnaviridae*) was present in the intestinal mucosa of UC patients which could impair the gut barrier, as was demonstrated in ex vivo and in vivo experiments[46]. Moreover, an increased faecal virome richness of eukaryotic viruses was found to associate with treatment failure of faecal microbiota transplantation (FMT) in patients with UC[42]. These findings combined suggest that a persistent eukaryotic virome expansion might underline the recalcitrant and refractory nature of IBD. In agreement with this hypothesis, our data also showed that the existing IBD therapeutics, biologics, immunosuppressants, glucocorticoids, and 5-ASA did not have a significant effect on changing the ileal abundance of *Torque teno viruses*. Therefore, the IBD therapeutic arsenal should be enriched with an additional treatment goal/endpoint of attenuation of the lingering eukaryotic virome bloom at the GI mucosa.

On the prokaryotic virus level, we found a broad depletion of bacteriophage taxa at the ileal mucosa in both CD remission and flare-up compared to HC, including those whose host were pathogenic bacteria: *Jedunavirus KpV80* (Host: *Klebsiella sp.*), *Bacteroides phage B124-14* (Host: *B. fragilis*), *Listeria phage BO25* (Host: *Listeria monocytogenes*), and *Streptococcus phage EJ-1* (Host: *Streptococcus pneumoniae*). In vitro studies of tissue culture cells have demonstrated that bacteriophages adhering to mucus defend against incoming pathogenic bacteria thereby protecting the underlying epithelium[6,7]. Therefore, the mucosal depletion of those bacteriophages whose host is pathogenic bacteria in CD may lead to an increase of their host bacteria in the gut. In favour of this assumption, *Klebsiella sp.*, *B. fragilis*, and *Listeria monocytogenes* were all reported to be elevated in the faeces of IBD patients[18,34,47]. Furthermore, we observed that the richness of lytic phages in the ileal mucosal virome was significantly diminished in CD compared to HC. These findings together suggest that mucosal bacteriophages may serve as 'sentinels' against the invasion of pathogenic bacteria, and the loss of these bacteriophages may keep the bacteriome unchecked, underlying the disease course of CD. Notably, we noticed a lower abundance of temperate phages than lytic phages (median: 22% vs. 78%) in the intestinal mucosa, falling within the 20% to 50% range for temperate phages in faecal phageome[48], despite at the lower end of the spectrum. Given that the ileal microenvironment is different from that in the lumen and faeces, phages at the ileal mucosa are kept in check simultaneously by mammalian epithelial cells, host bacteria, and stress molecules from both sides. Besides, lytic phages could attach to the mucosal layer of the intestine through its Ig-like domains on the capsid, thereby guarding against invasions of pathogenic bacteria and maintaining epithelial homoeostasis[6,7]. With all these factors at play, the mucosal phageome are anticipated to be different from the faecal phageome compositionally.

Concordantly, our study found that from an ecological standpoint, the phageome-bacteriome interaction network was substantially weakened in CD flare-up and remission compared to that observed in HC. However, the weakened phageome-bacteriome network was more severe in CD remission than in CD flare-up. These data suggest that the remission of intestinal inflammation in CD does not indicate the restoration of the microbiome (both phageome and bacteriome) at the intestinal mucosa; such prolonged, worsened phageome-bacteriome ecology during disease remission may consequently incur another episode (flare-up) of enteric inflammation in CD patients. A recent study reported that unstable bacteriome dynamics in quiescent CD significantly precede CD flares[49], again corroborating the importance of microbial ecology for intestinal homoeostasis in CD. Beyond bacteriome, our study also highlights the prominence of phageome in microbial ecology underlying the disease course. Hence, tailored therapies aiming to restore the phageome-bacteriome ecology (such as phage cocktail therapy) at the intestinal mucosa level are being proposed in the future for IBD treatment.

Our study also found that medications have significant impacts on the ileal virome composition, displaying both independent and interdependent effects with CD, as shown by both bioinformatic analysis and experiment. Amongst the medications for CD, biologics showed the largest association size with the ileal virome composition, followed by glucocorticoids, immunosuppressants, and 5-ASA. 5-ASA was recently demonstrated to be nullified by acetyltransferases expressed in the gut bacteria, causally leading to a diminished clinical efficacy in IBD treatment[50]. However, how the existing IBD medications coordinate with the collective microbiome to exert a clinical effect is largely unknown. In our study, though the virome-bacteriome network in CD was better restored by biologics compared to the other 3 medications, it still did not reach a level that was robustly similar to that in HC, further stressing a demand for novel therapies to target the virome-bacteriome ecology in CD.

Our study represents one of the first efforts to explore the enteric mucosal virome in both health and CD, particularly at the small bowel mucosa level. Notwithstanding this, the present study had a handful of limitations. First, unlike the well-established knowledge on the gut bacteriome, the gut virome is presently a critical 'dark matter' of the gut microbiome as the viral (especially prokaryotic viruses, namely bacteriophages) taxonomy and functionality are overall unclear (only 47.8% viral contigs could be taxonomically assigned to a known viral species in the present study), largely due the incomplete virome databases. Moreover, the genomic promiscuity, host jump events, and versatile lifecycles of bacteriophages that are intertwined with their bacteria host further complicate the effort to interrogate the virtual taxonomy and functionality of the large diversity of the gut phageome. Second, the RNA virome was not explored in this study, though prior studies have shown that it was substantially underrepresented in the faecal virome compared to the DNA virome[51]. Third, the majority of gut viruses (both eukaryotic viruses and bacteriophages) are to date not isolated or culturable. For instance, *Torque teno viruses* are recalcitrant to cultivation[37,52], which critically limits the downstream mechanistic interrogation of the individual roles of these viruses in disease. Fourth, considering the minuscule presence of viruses/phages in ileal biopsy specimens (as demonstrated by our nuanced VLPs staining by SYBR GOLD despite viral enrichment), a viral DNA amplification procedure was therefore conducted on extracted DNA prior to sequencing in this study; it may introduce bias towards identification of ssDNA viruses[53].

In summary, this trans-cohort study on the ileal virome of CD patients versus HC discovered a substantial richness depletion of bacteriophages and an expansion of eukaryotic viruses from the *Anelloviridae* family at the intestinal mucosa. Medications and diet had a crucial impact in shaping the gut mucosal virome and may partly play a role in restoring the virome in CD. Our observation of the persistent virome-bacteriome network disturbance at the ileal mucosa in CD remission following flare-up indicates that ecological restoration of the virome-bacteriome interplay should be considered as an additional

therapeutic endpoint in IBD, beyond the goal of a resolution of mucosal inflammation at the endoscopic and histologic levels.

## Methods

### Cohort description and inclusion criteria

Two cohorts of patients with Crohn's disease (CD) were enrolled in the present study, recruited respectively from two geographically distinct centres, the Six Affiliated Hospital of Sun Yat-sen University (Guangzhou, China) and The First Affiliated Hospital of Kunming Medical School (Kunming, China). Patients were diagnosed based on standard endoscopic, radiographical and histological criteria. A second group of age- and sex-matched healthy control participants (HC) was also included from the two centres, respectively, enrolled via the routine physical assessment regime. Overall, a total of 208 CD patients and HC were recruited (Table 1 and Supplementary Data 1): 51 CD patients and 51 HC in the Guangzhou cohort ($n = 102$) were recruited; 52 CD patients and 54 HC in the Kunming cohort ($n = 106$). The study was approved respectively by the Institutional Review Board (IRB) of the Research Ethics Committee of the Six Affiliated Hospital of Sun Yat-sen University (Ref. No: 2021ZSLYEC-245) and the IRB of Research Ethics Committee of the First Affiliated Hospital of Kunming Medical School (Ref. No: 2022.L.94). Inclusion criteria of CD patients included: having a clinical diagnosis of CD for more than 3 months and less than 3 years; agreeing to participate in this study; signing informed consent; and being able to complete environmental/disease/dietary questionnaire. Exclusion criteria of CD patients included: having used antibiotics/probiotics/prebiotics usage in recent 3 months; intestinal surgery (excluding individuals who had an appendectomy history of >3 years before the sampling date); infectious diseases; or cancer diagnosis. All HC underwent endoscopy assessment and were then included upon the confirmation that no inflammation/abnormality was found across the gut mucosal surface[54]. Participants of HC were excluded if they had any overt chronic diseases (including diabetes, hypertension, and autoimmune disorders), digestive illnesses or malfunctions (including abdominal distension, abdominal pain, and diarrhoea). All participants have provided written informed consent and consented to provide mucosal specimens of the terminal ileum via endoscopy. For participants under the age of 18, informed consent was obtained from their legally authorised representatives (LARs). Compensation was provided to all participants enrolled in our study. Sample saturation curve analyses based on the ileal mucosal virome at both the family and species levels showed that the current sample size ($n = 208$) of our study was sufficient to capture the overall virome diversity (Supplementary Fig. 1b), and hence allowed us to compare the configurational difference between HC and CD.

All participants completed the clinical survey (including disease course [remission versus flare-up], Harvey–Bradshaw Index[55], medications, intestinal inflammation assessment), metadata (including age, sex, anthropometric features [including height, body weight, BMI], geographical region, lifestyle) and dietary habit questionnaires (including market versus farm foods, consumption frequency of various food categories, all summarised in Supplementary Data 1). Clinical phenotypes and metadata of study participants was obtained by medical practitioners. Dietary questionnaire investigation was conducted by a dietitian. Procedures of endoscopic assessment, ileal biopsy obtainment, and sample storage followed a standardised operation procedure (SOP) shared between centres. To avoid centre- and batch-effect, all samples were transported to the Gastrointestinal Microbiome Laboratory at the Six Affiliated Hospital of Sun Yat-sen University, followed by virome enrichment and DNA extraction in one batch (detailed as below).

### Mucosal viral particles (VLPs) enrichment, DNA extraction, and sequencing

Virus-like particles (VLPs) were enriched from the ileal mucosal sample, refined from our previously described protocol[43,45,56]. Each of the terminal ileal biopsy was digested in digestive buffer (2 µl Collagenase D [Roche], 10 µl DNase I (10 mg/ml) [Roche], and 988 µl sterile saline magnesium buffer [SM buffer, 100 mM NaCl, 8 mM MgSO4, 50 mM Tris {pH 7.5} and 0.002% gelatin {w/v}]) at 37 °C for 1.5 h, with intensive vortex for every 20 min. 2/3 of the mixture was collected for viral DNA extraction, while the remaining 1/3 of the mixture was kept for bacterial DNA extraction. Biopsy suspension was centrifuged at $5000 \times g$ for 10 min to remove debris and cells, and the supernatant was then passed through a 0.22 µm filter to remove residual human and bacterial cells. The samples were then treated with lysozyme (pH 6.0–9.0) and incubated on a shaker at 37 °C for over 30 min, followed by chloroform treatment for 10 min at room temperature to degrade the remaining bacterial and human cell membranes. Then, non-membrane protected DNA was degraded with Baseline zero DNase (Epicentre) at 37 °C for 15–30 min, and DNA degradation was terminated with STOP buffer (30 mM EDTA) at 37 °C for 10 min. Viral particles (VLPs) were lysed with 3.8% SDS plus 100 µg/ml protease K at 56 °C for 40 min, and then treated with CTAB (2.5% CTAB supplemented with 0.5 M NaCl) at 65 °C for 10 min. Any remaining proteins were removed with chloroform treatment (equal volume as sample), followed by an equal volume of phenol:chloroform:isoamyl alcohol (PCI) treatment, and the nucleic acids were dissolved in the aqueous phase. The extracted DNA was purified and concentrated by DNA Clean & Concentrator kit (Zymo Research). The VLPs DNA was amplified for 4 h with Phi29 DNA polymerase (GenomiPhi V2 kit, GE Healthcare) before sequencing. Three independent reactions were conducted and then pooled together for DNA purification, for each sample. After quality control assessment by Agilent 5400, all qualified DNA samples were subjected to metagenomic library preparation using NEB Next Ultra DNA Library Prep Kit (New England Biolabs, USA, #E7370L), and all qualified libraries ($n = 208$) were sequenced on Illumina NovaSeq 6000 platform (Novogene, Beijing, China).

### Sequence processing and quality control

Raw reads were processed by *Trimmomatic* (v0.39)[57] for quality control, trimming adaptors, removing low-quality reads or those with multiple N bases (below quality 3), and sequences less than 50 bp in length. Human reads contamination was removed by aligning to the human genome reference database (GRCh38 p12) via *KneadData* (v0.7.4)[58]. A total of 4532 million paired clean reads were obtained for contig assembly.

### Viral contig assembly and identification

Clean reads were assembled using *MEGAHIT* (v1.2.9)[59], a widely used next-generation sequencing (NGS) assembler for its ultra-fast and memory-efficient characteristics[60]. Following contig assembly, *CD-HIT* (v4.7) was used to remove redundant contigs which shared over 90% sequence similarity with other contigs[61]. Those non-redundant contigs ≥1.5 kb were piped through *VirSorter2* (v2.2.3)[62], *DeepVirFinder*[63], and *CAT* (v5.2.3)[64] for viral contigs identification. Those Contigs ≥5 kb or contigs ≥1.5 kb and circular (circular viral contigs were identified using *CheckV* (v1.0.1)[65]) were then sorted using *VirSorter2* and/or *DeepVirFinder* (based on the criteria of score ≥0.7 and $p < 0.05$ in *DeepVirFinder*), in order to discern different viral categories while excluding those falsely identified as RNA virus. Of those contigs, the ones with *VirSorter2* score >0.5 and *DeepVirFinder* score ≥0.8 and $p < 0.05$, or those with *VirSorter2* score >0.9 and *DeepVirFinder* score ≥0.7 and $p < 0.05$, were classified as candidate contigs of viral origin. The remaining contigs were then processed using *CAT*, and those contigs classified as eukaryotic viruses by *CAT* were also considered to be of viral origin.

To remove bacteria-sourced false positive contigs, we evaluated the bacterial gene enrichment in each contig by assessing the number of hits to bacterial universal single-copy orthologs (BUSCO)[66]. For some of bacterial BUSCO genes can also be present in viral genomes[67], we assessed the BUSCO genes (bacteria_odb10, from BUSCO database

[https://busco.ezlab.org/]) present within viral genomes (viral_refseq_v212) using *BUSCO* (v5.3.2)[68], and obtained a range of BUSCO ratio value (the rate of BUSCO hits per total number of genes in each viral RefSeq genomes[69]) of 0–0.01. We then assessed the BUSCO ratios of each viral contig, and compared them to the viral RefSeq BUSCO ratio values. For further evaluating the viral genes enrichment, we performed an *hmmsearch*[70] of all contigs against the viral protein family modules (VPFs)[71], with hits defined as an *e*-value ≤ 0.05. Only those contigs with a BUSCO ratio ≤0.01 or a BUSCO ratio >0.01 or at least 3 VPF hits were finally identified as viral contigs. The remaining contigs, which were identified as known ssDNA viruses and shared ≥95% nucleotide identity across viral genomes by *CAT*, were also considered as viral contigs[69]. Ultimately, 3.01% of the assembled, non-redundant contigs were identified as viral contigs.

Open Reading Frames (ORFs) were predicted and identified using *Prodigal* (v2.6.3)[72], and the amino acid sequences with a minimum length threshold of 100 amino acids were selected for further analyses. Viral contigs >10 kb in length and identified as double-stranded (ds) DNA were clustered and assigned to a known viral taxonomy with RefSeq-v211 using the pipeline *vContact2* (v0.11.3)[73] based on *diamond* (v2.0.11)[74]. The remaining predicted amino acid sequences of ORF were then matched against the standard subset of the RefSeq-v212 database, which contains only viral reference proteins, using *blastp* (v2.12.0, *e*-value < $10^{-5}$)[75] as previously reported[43,76]. Each contig was assigned a taxonomy using a voting system for virus taxonomy assignment at different taxonomic levels[76,77]. The voting system annotated each viral contig with the best-hit viral taxa, then compared the taxonomic assignments of ORFs within the contigs of interest, and annotated the contigs based on the majority ORF assignment. Contigs without a majority ORFs taxonomic assignment among amino acid sequences were then compared at a higher taxonomic level, until majority taxonomic assignment was obtained. Besides, contigs with less than one ORF per 10 kb were not taxonomically annotated, for they had limited similarity with the known viruses[77]. For the remaining viral contigs, which were identified as eukaryotic or ssDNA viruses, *CAT* (v5.2.3) was used to assign the viral taxonomy.

Viral abundance was calculated with *Bowtie2* (v2.3.5.1)[78] by aligning clean reads to curated viral contigs, and the sequence counts were standardised to RPKM (reads per kilobase million) using *BBMap* (v38.84) (*BBMap*—Bushnell B.—sourceforge.net/projects/bbmap/). These values were used to generate a viral abundance table at different taxonomic levels. The flow chart of analysis was summarised in Supplementary Fig. 1a.

### Virome function analysis
Viral contigs were translated into amino acid sequences and their respective Coding Sequences (CDS, the nucleotide sequence which directly encoded the amino acid in a protein) using *Prokka* (v1.12)[79]. Virome functions were predicted by annotating all viral contig-derived amino acid sequences via *diamond* (v2.0.11)[74] and *blastp* (v2.12.0), aligning against the viral protein sequence and functional modules of the Uniprot knowledgebase (release 2022_04)[80]. Predicted functions were classified and binned based on Gene ontology terms[81,82] and Pfam protein family identities[83], and abundance values were expressed in RPKM via *Bowtie2* alignment and *BBMap* transformation. To establish the presence and absence of viral functions, only the top 250 abundant proteins and their function modules were taken into further analysis.

### Bacteriophage lifestyle prediction and bacterial host prediction
The lifestyle (lytic versus temperate) of bacteriophages was predicted by *PhaTYP*[84], a bacteriophage lifestyle prediction pipeline based on deep learning. The lifestyle categories of the viral contigs with a confident prediction score >0.5 were used for further analysis.

The bacterial host of bacteriophages of interest was identified by searching the Virus-Host DB (https://www.genome.jp/virushostdb/ note.html)[85] and the result was shown in Supplementary Data 2. Moreover, we conducted virus/bacteriophage host prediction using the *CHERRY* algorithm (prediction score >0.9) in the pipeline *PHYBOX* (https://phage.ee.cityu.edu.hk/)[16], a state-of-the-art pipeline for bacteria host prediction of bacteriophages based on CRISPR spacer alignment and machine learning model.

### Mucosal bacterial DNA extraction and 16S rDNA sequencing
In parallel to viral DNA extraction, bacterial DNA extraction was performed on the same intestinal biopsy (the remaining 1/3 mixture of the biopsy preparation during the virome DNA extraction procedure). Given the previous report that acute mechanical cell disruption (bead-beating) had no advantage in bacterial DNA extraction from intestinal biopsies[86], we chose gentle mechanical disruption coupled with chemical and enzymatic lysis in our mucosal bacterial extraction protocol. Biopsy suspension was centrifuged at $100 \times g$ for 1 min to remove large debris and tissue, and the supernatant was treated with Tris-EDTA buffer at 68 °C for 10 min to cease DNase digestion. Each sample was then added with lysozyme, incubated in the shaker (200 rpm) at 37 °C for 90 min, and intermittently vortexed every 20 min. The supernatant was transferred to the Zymo-Spin™ III-F filter in a collection tube and centrifuged at $8000 \times g$ for 1 min. Once all supernatant was passed through the filter, 1200 μl of Genomic Lysis Buffer (Zymo Research, D3004-1-100) was added to the filtrate in the collection tube and mixed well. Then, all mixture was transferred to Zymo-Spin™ IICR Column, and centrifuged at $10,000 \times g$ for 1 min. The DNA collected in the Zymo-Spin™ IICR Column was then washed with Pre-Wash Buffer (Zymo Research, D3004-5-15) and g-DNA Wash Buffer (Zymo Research, D3004-2-50), and eluted with DNA Elution Buffer. The DNA preparations extracted from the ileal mucosa were then subjected to 16S rDNA sequencing (V4 region, PE300) on the MGISEQ-2000 (BGI, Shenzhen, China) platform.

### Bacterial 16S rDNA sequencing data analysis
Bacterial 16S rDNA sequencing results were analysed by *Qiime2* (v2022.2.0)[87] as follows: (1) denoising including reads trimming, quality filtering, and chimaera removal; (2) sequences dereplication (sequences clustering performed at 100% identity); (3) feature table establishment; (4) taxonomy classification based on the SILVA SSU database (SILVA 138 SSU Ref NR 99)[88]. Ultimately, a bacterial composition table was generated for further analysis.

### Metadata categories and covariates with virome/bacteriome variation
All metadata factors were classified into seven categories including anthropometrics (height, body weight, and BMI), general metadata (age, sex, smoker, animal contact, and other lifestyle factors), geography (Guangzhou versus Kunming), disease (HC versus CD), intestinal inflammation (Non-inflammation versus CD remission versus flare-up), co−morbidities (hypertension, allergic disease and appendectomy), medications (including glucocorticoids, 5-aminosalicylic acid [5-ASA], immunosuppressants, and biologics, Chinese medicine, antacid drug and hypotensive drugs in the past 3 months), and dietary habit (such as processed meat, fruits and vegetables) (Supplementary Data 1).

Covariates of virome/bacteriome were identified by calculating the associations between all host factors from metadata (including continuous, categorical, and logical variables) and viral/bacterial compositional data, using *envfit* function in the vegan R package (999 permutations with a false discovery rate of FDR < 0.05)[89,90]. *Envfit* function performed linear correlations and *MANOVA* analysis for continuous and categorical variables, respectively. The combined effect size of host factors when categorised into the redefined seven categories was estimated with variance partitioning analysis function[91] via the *vegan* package in R (999 permutations; R-square was adjusted by number of observations, and number of degree of freedom in the fitted model[92]).

## Microbiome data analysis

The virome and bacteriome abundance tables were transformed into relative-abundance tables and used for the following analysis. Multi-omics variations and the sources of variances in the prokaryotic and eukaryotic virome and the bacteriome were investigated in conjunction with the patients' metadata, by the pipeline Multi-Omics Factor Analysis tool (MOFA)[14]. The α diversity metrics (richness, evenness, Simpson and Shannon index) were calculated using *vegan* package in R. The overall compositional differences between HC and CD (including remission versus flare-up) were evaluated using the Bray–Curtis dissimilarity metrics, and visualised in PCoA plot using *amplicon* and *ggplot2* packages in R. The viral-bacterial α diversity correlations were calculated and visualised using R via *ggcor*, while viral-bacterial abundance correlations were performed using *FastSpar* (v1.0.0)[93]. Redundancy analysis (RDA) for explaining relationships between grouping factors (HC versus remission versus flare-up) and relative abundance of viral taxa was performed using *vegan* package in R[94]. Feature taxa associated with metadata variables (HC versus CD, intestinal inflammation, medications, and dietary factors) were identified using *MaAsLin2* package in R[95]. These results were visualised (boxplot, violin plot, column plot, volcano plot, point plot and jitter plot) using *ggplot2* package in R, while heatmaps were generated using *ComplexHeatmap* package in R. Venn plots were created with *Evenn* (http://www.ehbio.com/test/venn/#/)[96].

## Mucosal VLPs preparation from human patient samples for animal gavage

Mucosal VLPs were isolated and enriched from surgically resected ileum specimens from patients with CD ($n = 4$) or colorectal cancer (as non-CD controls, $n = 4$). Patients' samples were collected from Department of Gastrointestinal Surgery, The Sixth Affiliated Hospital, with informed consent from each patient in accordance with ethical regulations. Fresh ileal resections were collected under the guidance of a dedicated GI surgeon, and the inflammation in CD ileal resection samples were confirmed and assessed by a clinical pathologist. Non-CD controls included patients undergoing partial ileal resection and ascending colectomy for colorectal cancer, and normal histological features of the ileal resection samples were confirmed by a clinical pathologist. In total, 100 mg mucosal layer of ileal resection samples were gently scraped with sterile scalpel, then washed with 500 ul 0.02 μm-filtered SM buffer, and centrifuged at $700 \times g$ for 3 min to remove large organic particles, followed by centrifuging at $3200 \times g$ for 30 min to separate bacteria from viruses. The viral supernatants were then passed through a 0.45 μm filter and a 0.22 μm filter to remove the remaining bacteria. Following that, viral supernatants were then up-concentrated with a 50 kDa Centriprep® filter to remove residual metabolites.

The viral preparations were then VLP-quantified[97]. The viral supernatant was serially diluted by 10-fold followed by fixation in a 0.02 μm Anodisc filter (Cytiva, 6809-6002). VLPs were then stained in 0.02 μm-filtered 1x SYBR GOLD for 15 min, and then washed with 1 ml 0.02 μm-filtered Milli-Q Water. The images of stained VLPs were captured by Leica TCS SP8 confocal microscope (100x). Particles with a diameter of less than 0.5 μm in diameter were regarded as VLPs and were then enumerated. SM buffer was subjected to the same procedure as a negative control. Five images were captured for each preparation (Supplementary Fig. 14a).

## Mice

6-8 week-old male C57BL/6J mice used in our research were all purchased from Beijing Vital River Laboratory Animal Technology Corporation (Beijing, China). Mice were housed in standard cages (temperature: 18–25 °C; humidity: 50–60%; 14-h light/10-h dark cycle) in Sun Yat-sen University or The Six Affiliated Hospital of Sun Yat-sen University for different batches of animal experiments, and the procedures were conducted in accordance with the animal use protocol and animal ethical regulations approved by the Institutional Animal Care and Use Committee (IACUC) of Sun Yat-sen University (No. 2023001439) and The Six Affiliated Hospital of Sun Yat-sen University (IACUC-2021101101). All mice were co-housed for 1 week to homogenise the microbiome before the beginning of each animal experiment.

For DSS-induced acute intestinal inflammation model, age- and weight-matched mice were randomised and allocated to non-CD-VLPs-gavage, CD-VLPs-gavage groups and placebo (SM buffer, without VLPs) group (referred to as non-CD, CD and placebo groups respectively; $n = 10$ mice in each group). The timeline and the schematic were summarised in Fig. 7a and Supplementary Fig. 15a. Ileal VLPs derived from non-CD and CD patients were enumerated by SYBR GOLD staining and diluted to the concentration of $2 \times 10^9$ VLPs/ml with SM buffer. Before VLPs administration, the gastric acid of mice was neutralised by intra-gastric gavage of 100 μl 1 M NaHCO$_3$, and 15 min later the mice were administered non-CD or CD VLPs by oral gavage (200 ul each mouse for about $4 \times 10^8$ VLPs). Overall, mice were administered human ileal VLPs gavage every other day for a total of 6 times until 8 weeks of age. After VLPs gavage for 2 weeks, all mice were treated with 2% DSS for 6 days, followed by a 4-day DSS/washout period. Before scarifice (on the third day of DSS/washout period), all mice underwent endoscopic evaluation, and then sacrificed 1 day after. Disease Activity Index (DAI) was evaluated on the day of sacrifice, including measurement of body weight loss (0: <1%, 1: 1–5%, 2: 5–10%, 3: 10–20%, 4: >20%), stool consistency (0: normal, 2: loose stools, 4: diarrhoea), and rectal bleeding (0: negative, 2: occult bleeding, 4: gross bleeding). Colonic tissues and faecal content of all mice were collected. Throughout the experiment, all mice were treated with the same sterilised diet and water.

For 2,4,6-trinitrobenzene sulfonic acid (TNBS)-induced mouse model with acute intestinal inflammation, mice were treated according to a previous report[98]. Age- and weight-matched mice were randomised into non-CD-VLPs-gavage and CD-VLPs-gavage groups, and orally administered human ileal VLPs every other day for 6 times. The timeline and the schematic were summarised in Supplementary Fig. 16. On day 5, mice were immunised with 150 μl of the TNBS pre-sensitisation solution (The TNBS pre-sensitisation solution was prepared as follows: Acetone and olive oil [MAKLIN, O815211] were mixed in a 4:1 volume ratio by vortexing rigorously, followed by adding 1 volume of 5% [w/v] TNBS [Sigma, p2297] solution to 4 volumes of acetone/olive oil mixture by vortexing rigorously) on the shaved skin area. After 1 week, on day 12, 100 μl of 2.5% (w/v) TNBS, which was diluted in 50% ethanol, was administered intrarectally through a catheter into the colon for 4 cm, and all mice were then sacrificed 4 days afterwards. Colonic tissues and faecal content of all mice were collected, and DAI score was evaluated as above.

## RNA isolation, quantitative RT-PCR and transcriptomics sequencing

Total RNA was extracted from mouse colonic mucosa using TRIzol reagent (Invitrogen, 15596026) and then used for cDNA synthesis with HiScript III RT SuperMix for qPCR (Vazyme, R323-01). Quantitative PCR was performed to quantify the gene expression of pro-inflammatory cytokines and chemokines using 2×Hieff® qPCR SYBR Green Master Mix (No Rox) in QuantStudio 7 Flex instrument (Applied Biosystems). In addition, transcriptomics sequencing (RNA-seq) was performed with an Illumina Novaseq 6000 platform (PE150, Magigene, Guangdong, China), and genes were annotated with HISAT2[99] by mapping to the Mus musculus genome (assembly GRCm39). Differentially expressed gene (DEG) analysis was performed with DESeq2[100], and the genes that fulfilled the criteria FDR adjusted $p < 0.05$ and fold change >2 were considered to be differentially expressed. All of the differentially expressed genes

were utilised as the input for Gene Set Enrichment Analysis (GSEA) and Gene Ontology analysis (focus on biological process), and were visualised with *ggplot2* R package. The primer pairs for quantitative RT-PCR are shown in Supplementary Table 1.

## Mouse gut virome and bacteriome interrogation

The same faecal pellet of each mouse was cut into 2 pieces; the first half was used for virome DNA extraction and virome analysis, while the second half was used for metagenomic DNA extraction and bacteriome analysis. For virome DNA extraction, the faecal content was suspended in SM buffer followed by VLPs isolation, DNA extraction, virome sequencing and data analysis, according to the methods mentioned above. For mouse virome data analysis, the trimmed and qualified reads were aligned to our in-library viral contigs constructed in humans as described above. For bacteriome interrogation, total faecal DNA was extracted and purified using Quick-DNA faecal/Soil Microbe Kits (Zymo Research, D6010) according to the manufacturer's protocol. After quality control assessment by Agilent 5400, all qualified DNA samples were subjected to metagenomic library preparation using NEB Next Ultra DNA Library Prep Kit for Illumina (New England Biolabs, USA, #E7370L), and all qualified libraries ($n = 40$) were sequenced on Illumina NovaSeq 6000 platform (Novogene, Beijing, China). For metagenomic data analysis (the bacteriome part), raw reads were processed by *Trimmomatic* (v0.39)[57] for adaptors trimming and low-quality reads removal. Human reads contamination was removed by aligning to the human genome reference database (GRCh38 p12) via *KneadData* (v0.7.4)[58]. Overall, an average of 45.27 million ± 0.34 million (mean ± S.E.) clean reads for each sample were obtained for further analysis. Bacteria taxonomy was profiled using *MetaPhlAn4* (v4.0.4)[101] with default settings (min_map_q_val: 5; stat_q: 0.2; read_min_len:70).

## In vitro and in vivo experiments investigating the effect of medications on the gut virome

To investigate the impact of drugs on the intestinal virome in vitro, we cultivated human microbiome preparations with medications of interest. We collected faecal samples from 5 healthy participants and dissolved them in a reductive 10% Brain Heart Infusion (BHI) medium with a vortex. Subsequently, the suspension was centrifuged at a speed of $700 \times g$ to remove large particulates from the faeces. The suspension consisting of bacteria and viruses, was cultivated with 200 mM of 5-Aminosalicylic Acid (5-ASA), 50 uM of Azathioprine (AZA), 100 uM of Methylprednisolone (MP), or a negative control (10% DMSO), under anaerobic conditions for 12 h. Following the co-culture, viral DNA was extracted and the target bacteriophages were quantified by qPCR, using primers (shown in Supplementary Table 1) designed from the in-house assembled virome contigs from our human virome dataset. For each test, 1 ng of DNA was used as the total input for assessing the relative abundance of the target bacteriophages.

For the in vivo experiment exploring the effect of medications on gut virome, mice were administered each medication of interest every other day for a total of 6 times. These medications included 5-ASA (100 mg/kg/day), AZA (1.5 mg/kg/day), and MP (10 mg/kg/day). The faecal samples were collected prior to (Day 9) and post the administration of medication (Day 21) (shown in Supplementary Fig. 11a). Subsequently, virome DNA extraction and quantification of target phages were performed as described above.

## Reporting summary

Further information on research design is available in the Nature Portfolio Reporting Summary linked to this article.

## Data availability

The raw sequence data reported in this paper deposited in the Genome Sequence Archive (Genomics, Proteomics & Bioinformatics 2021) in National Genomics Data Centre (Nucleic Acids Res 2022), China National Centre for Bioinformation/Beijing Institute of Genomics, Chinese Academy of Sciences, under the Bioproject number PRJCA018565 (GSA-Human: HRA005245, https://ngdc.cncb.ac.cn/gsa-human/submit/hra/subHRA007362/; GSA: CRA012045, https://ngdc.cncb.ac.cn/gsub/submit/gsa/subCRA018827/ and CRA011968, https://ngdc.cncb.ac.cn/gsub/submit/gsa/subCRA018805/). In addition, the following public databases were also used in this paper: Virus-Host DB (https://www.genome.jp/virushostdb/note.html), Uniprot knowledgebase database (release 2022_04, https://www.uniprot.org/help/uniprotkb), SILVA SSU database (SILVA 138 SSU Ref NR 99, https://www.arb-silva.de/), RefSeq-v212 database (https://www.ncbi.nlm.nih.gov/refseq/), and BUSCO database (bacteria_odb10, https://busco.ezlab.org/) and human genome reference database (GRCh38 p12, https://www.ncbi.nlm.nih.gov/datasets/genome/). Source data are provided with this paper.

## Code availability

The code utilised in this study has been made publicly available at the following GitHub repository: https://github.com/ouczt/Crohn_disease_virome_Zuotao_Lab. It can also be accessed via https://doi.org/10.5281/zenodo.10538381.

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

## Acknowledgements

This work was jointly supported by the National Natural Science Foundation of China (NSFC grant Nos. 82172323, 32100134 and 82060107), the Municipal Key Research and Development Program of Guangzhou (grant No. 202206010014), a seed fund from the Sixth Affiliated Hospital of Sun Yat-sen University and Sun Yat-sen University (2022JBGS03), and an Applied Basic Research Projects of Yunnan Province (202201AW070019). Figure 1a was created by *mapdata* R package and BioRender.com, and Figs. 1b and 7a and Supplementary Figs. 10a, 11a, c, e, 15a, 16a were created by BioRender.com.

## Author contributions

TZ devised the study. ZRC performed data analysis and drafted the manuscript. DJF and YS conducted subject recruitment, collected clinical samples and data. ZYH, YL, FZ and RPS assisted in viral and bacterial DNA sample preparation. YS, HJY and YLM assisted in sample collection and questionnaire investigation. PL and QL provided significant intellectual contribution to the manuscript. TZ and XJW designed the study, supervised the study and revised the manuscript.

## Competing interests

The authors declare no competing interests.

## Additional information

[1]Key Laboratory of Human Microbiome and Chronic Diseases (Sun Yat-sen University), Ministry of Education, Guangzhou, Guangdong, China. [2]Guangdong Institute of Gastroenterology, The Sixth Affiliated Hospital of Sun Yat-sen University, Sun Yat-sen University, Guangzhou, Guangdong, China. [3]Centre for Faecal Microbiota Transplantation Research, The Sixth Affiliated Hospital of Sun Yat-sen University, Sun Yat-sen University, Guangzhou, Guangdong, China. [4]Biomedical Innovation Centre, The Sixth Affiliated Hospital, Sun Yat-sen University, Guangzhou, Guangdong, China. [5]Department of Gastrointestinal Endoscopy, The Sixth Affiliated Hospital of Sun Yat-sen University, Sun Yat-sen University, Guangzhou, Guangdong, China. [6]Guangdong Provincial Key Laboratory of Colorectal and Pelvic Floor Diseases, The Sixth Affiliated Hospital, Sun Yat-sen University, Guangzhou, Guangdong, China. [7]Gastroenterology, The First Affiliated Hospital of Kunming Medical University, Kunming, Yunnan, China. [8]Yunnan Province Clinical Research Centre for Digestive Diseases, Kunming, Yunnan, China. [9]Yunnan Geriatric Medical Centre, Kunming, Yunnan, China. [10]Department of Gastroenterology, The Sixth Affiliated Hospital, Sun Yat-sen University, Guangzhou, Guangdong, China. [11]Department of Food Science and Engineering, College of Life Science and Technology, Jinan University, Guangzhou, China. [12]Department of Colorectal Surgery, The Sixth Affiliated Hospital of Sun Yat-sen University, Sun Yat-sen University, Guangzhou, Guangdong, China. [13]These authors contributed equally: Zhirui Cao, Dejun Fan, Yang Sun. ✉e-mail: sunyang_doctor@vip.sina.com; wuxjian@mail.sysu.edu.cn; zuot@mail.sysu.edu.cn

