## [Peer Review File · Nature Communications]

REVIEWER COMMENTS

Reviewer #1 (Remarks to the Author):

The authors of the manuscript entitled “The gut ileal mucosal virome is disturbed in patients with Crohn’s disease and 2 exacerbates intestinal inflammation in mice” elegantly described the virome variation in a large cohort of CD patients enrolled in China and sought to correlate the virome dysbiosis in association to that of the bacteriome and clinical metadata.

The manuscript is well-written and the results well-described. However, I have some concerns:

Considering the vast amount of data and the precious availability of clinical information along with three layers of omics profiling (i.e., bacteriome, eukaryotic and prokaryotic virome), why did not the authors try to stratify patients by using machine-learning approaches to simultaneously analyze the different characteristics, pinpointing, as a result, the different omics factors associated with specific patients’ metadata? In this regard, I would suggest MOFA as a possible multi-omic analysis tool.

By this approach, the authors could shed light on, for example, the different microbiota characteristics in association with dietary habits and/or response to drugs, among others. Moreover, MOFA could explain the source of the variance found in the virome and bacteriOME in the CD cohort.

The authors often use terms such as “impact”, and “influence”. However, I believe that, in this context, they found an association between clinical characteristics and microbiota dysbiosis, not proving any causal link between them. So, I would suggest that the authors should mitigate the wording when necessary.

My major concern is the in vivo study. The authors sought to demonstrate that virome-derived particles may worsen colitis. However, they used the DSS model, which, to the best of my knowledge, is a colonic inflammation model and not an experimental approach properly mimicking CD, for which the TNBS model is more adequate.

Therefore, I suggest that they should repeat the experiment with the TNBS, and try to replicate the results.

Also, I propose to perform in vivo experiments with different sets of VLPs derived from different types of patients (different drug regimens, different extent of inflammation, decided based on their computational results), to prove the causal link (if any) they proposed in the computational approach.

The DSS-induced colitis is normally very strong with robust evidence of colonic ulcerations, that, in my opinion, are not appreciable in Figure 7. In this regard, I suggest better images and indicate the Disease Activity Index as a combination of the different parameters to be evaluated in experimental models of intestinal inflammation.

Reviewer #2 (Remarks to the Author):

The manuscript titled "The gut ileal mucosal virome is disturbed in patients with Crohn's disease and exacerbates intestinal inflammation in mice," submitted by Zhirui et al., provides a comprehensive description of the ileal tissue virome landscape from patients with Crohn's disease (CD) in clinical remission or flare-up, and analyzes virus-bacteria interactions. This analysis utilizes data from two geographically distinct cohorts and includes extensive metadata information. Some revisions are necessary before publication.

Major Points:

1. Given that tissue virome data was utilized, it is assumed that most of the data belong to the host. Could the authors quantify the proportion of virome data? This would aid in evaluating the quality of the virome data.
2. The abundance of lytic phages is notably higher than that of temperate phages in this study, contrary to previous understanding in stool (R. Sausset et al., 2020). This difference may be attributed to the methodology employed to predict the phages' lifestyles. Or is it because of the ileal environment? Can the authors provide a more detailed discussion on this?
3. Figure 1E is unnecessary, as it has been reported many years ago.
4. The authors assert that geography has a greater impact on the bacterial community compared to inflammation. Further information is needed regarding the bacterial community. In patients with IBD, the expansion of Proteobacteria has been widely reported. What major differences exist between Guangzhou and Kunming?
5. Line 243: The Shannon diversity of the ileal virome was not significantly reduced in the CD flare-up group compared to the HC group in the Guangzhou cohort (Fig 2D).
6. Line 279: Please clarify the percentage of virome data that can be assigned at the species level.

7. Line 424: Why is the abundance of a single bacterium correlated with numerous phage species with a similar correlation coefficient? Could the authors reevaluate the raw data? Please refer to a classic publication (Norman 2015).

8. Line 578: A significant point is the absence of experiments on DSS-induced mice without any VLP treatment. This omission hinders a deeper understanding of the data.

9. The discussion presented requires further elaboration. It is advisable to delve into previous publications that encompass both stool and tissue studies on IBD, in order to contextualize and enhance the depth of the current findings. Please refer to some publications (Wagner 2013; Norman 2015; Clooney 2019; Liang 2020,2021;).

Minor Points:

Line 205: Mislabeled? Figure 1G&I/1J?

Line 396: host -> hosts

Line 541: should be "associated with"

Numerous grammar errors or typos, especially in the Methods section, require revision. Please carefully edit the manuscript.

Point-to-point response to the reviewers' comments

Reviewer #1 (Remarks to the Author):

The authors of the manuscript entitled "The gut ileal mucosal virome is disturbed in patients with Crohn's disease and 2 exacerbates intestinal inflammation in mice" elegantly described the virome variation in a large cohort of CD patients enrolled in China and sought to correlate the virome dysbiosis in association to that of the bacteriome and clinical metadata.

Response: We appreciate the reviewer's commendation.

Major points

1-The manuscript is well-written and the results well-described. However, I have some concerns: Considering the vast amount of data and the precious availability of clinical information along with three layers of omics profiling (i.e., bacteriome, eukaryotic and prokaryotic virome), why did not the authors try to stratify patients by using machine-learning approaches to simultaneously analyse the different characteristics, pinpointing, as a result, the different omics factors associated with specific patients' metadata? In this regard, I would suggest MOFA as a possible multi-omics analysis tool. By this approach, the authors could shed light on, for example, the different microbiota characteristics in association with dietary habits and/or response to drugs, among others. Moreover, MOFA could explain the source of the variance found in the virome and bacteriome in the CD cohort.

Response: We appreciate the reviewer's insightful suggestion to add such a useful multi-omics analysis tool (MOFA)¹ to improve our analysis. We conducted this analysis and presented the data and findings in **lines 206-218** in the manuscript along with the newly added figures **Supplementary figure 3C-F**.

The findings are also briefed as below:

Herein, we investigated both the variations and the sources of variances in the multi-omics data (prokaryotic and eukaryotic virome, and bacteriome), by MOFA. The virome collectively explained 30.5% of the overall variation in the multi-omics data, whilst the bacteriome explained 39.5% of the overall variation (**Supplementary figure 3C**). Furthermore, the prokaryotic virome had a larger weight in explaining the variations compared to the eukaryotic virome (**Supplementary figure 3E**). These findings together suggest that the virome (particularly prokaryotic virome) had a nearly comparable explanatory power with the bacteriome in the total variation of mucosal microbiome in CD.

Subsequently, through MOFA analysis, we identified 6 predominant microbiome factors that captured the most varying characteristics within the multi-omics mucosal microbiome dataset (differentially contributed by the virome and bacteriome variances, shown in **Supplementary figure 3C**; further detailed contributions by prokaryotic and eukaryotic virome, shown in **Supplementary figure 3E-F**), and then pinpointed the associations between patients' metadata and each microbiome factor (data shown in **Supplementary figure 3D**). The results showed that the microbiome factors 2 and 4, which captured

microbial variations majorly sourced from virome, were significantly associated with CD-related phenotype (including CD vs. HC, and intestinal inflammation) (**Supplementary figure 3D**). In contrast, the microbiome factor 1, which predominantly captured microbial variations majorly sourced from bacteriome, had robust associations with geography- and dietary habits-related factors in patients (**Supplementary figure 3D**). These data together indicate that different microbiome components in the intestinal mucosal microbiome are differentially influenced by host factors. The mucosal virome is more associated with host-intrinsic factors (intestinal inflammation), whilst the mucosal bacteriome is largely associated with host-extrinsic factors (geography and diets).

[1] Argelaguet R, et al. Multi-Omics Factor Analysis-a framework for unsupervised integration of multi-omics data sets. *Mol Syst Biol* 14, e8124 (2018).

2-The authors often use terms such as “impact”, and “influence”. However, I believe that, in this context, they found an association between clinical characteristics and microbiota dysbiosis, not proving any causal link between them. So, I would suggest that the authors should mitigate the wording when necessary.

Response: We thank the reviewer for this very wise suggestion. We have thoroughly gone through the manuscript, and tuned down our expressions to avoid any conclusive or causal wordings where only associations were indicated. Such amendments are highlighted in red in the revised manuscript.

3-My major concern is the *in vivo* study. The authors sought to demonstrate that virome-derived particles may worsen colitis. However, they used the DSS model, which, to the best of my knowledge, is a colonic inflammation model and not an experimental approach properly mimicking CD, for which the TNBS model is more adequate. Therefore, I suggest that they should repeat the experiment with the TNBS, and try to replicate the results.

Response: We concur with the viewpoint of the reviewer, and we therefore newly added a TNBS animal model in this revision, as a new model as per your suggestion, to investigate whether the findings of CD mucosal virome eliciting intestinal inflammation could be replicated in this new *in vivo* model (**Supplementary figure 16**). The results are now shown in **lines 630-633** and in the newly added figures **Supplementary figure 16A-H**. The results are also briefly shown as below. Overall, the findings in the TNBS model were in agreement with our findings in the DSS model, both showing that CD ileal virome evoked a more pronounced intestinal inflammation in mice, compared to non-CD ileal virome.

Our findings in TNBS mice are briefed as follows:

In comparison to mice administered with non-CD virome particles, mice administered with CD ileal virome particles exhibited more pronounced intestinal inflammation after TNBS treatment, as evidenced by lower survival rate (*Log-rank* test $p=0.07$, 90% in non-CD group versus 70% in CD group, **Supplementary figure 16B**), shortened colon lengths, elevated disease activity index (DAI), and higher histological inflammation scores (*Mann-Whitney*

tests, $p=0.0045$, $p=0.0324$, $p=0.16$, respectively, **Supplementary figure 16C-G**). Furthermore, we observed increased mRNA expression levels of pro-inflammatory cytokines in mice administered with CD ileal virome compared to those administered with non-CD ileal virome, including IL-1 α , IL-1 β , and IL-6 (*Mann-Whitney* test, all $p<0.05$, **Supplementary figure 16H**). In summary, our concurrent findings between the TNBS mouse model and the DSS mouse model validated the pro-inflammatory role of the ileal mucosal virions derived from CD patients in worsening intestinal inflammation, hence establishing a causal link.

4-Also, I propose to perform *in vivo* experiments with different sets of VLPs derived from different types of patients (different drug regimens, different extent of inflammation, decided based on their computational results), to prove the causal link (if any) they proposed in the computational approach.

Response: We greatly appreciate the reviewer's suggestion and believe such additional experiments would substantially improve our study. In our computational, bioinformatics analyses, we observed significant associations between different drug uses, different extent of inflammation (flare vs. remission), and the altered mucosal virome composition. In this revision, we managed to explore the causal relationship between different drug intakes and intestinal virome, by both *in vitro* and *in vivo* models. However, for *in vivo* studies to explore the causal relationships between the mucosal VLPs from patients with varying extent of inflammation and intestinal inflammation in mice, we practically rely on sufficient attainment of VLPs from surgically resected mucosal specimens from patients, as opposed to endoscopy-derived mucosal specimens; however, not all patients are eligible for surgery and only those **in clinical remission** are subject to surgical sample collection in clinical setting. Therefore, it unfortunately precludes us from getting sufficient amount of VLPs from all patients, along the whole inflammation severity spectrum, for *in vivo* gavage and downstream causal relationship interrogation. In light of this limitation, we only investigated the causal versus consequence relationship between drug intakes and intestinal virome, and the data are now shown in **lines 505-530** and in the newly added figures **Supplementary figure 10 & 11**.

Our findings are also briefed as below:

To explore the causal versus consequence relationship between CD medications and gut bacteriophages, we harnessed an ***in vitro* reductionist model** where we incubated faecal microbiota preparations from 5 healthy individuals with different medications of interest (5-ASA, AZA, and MP, respectively), and then specifically probed the abundance changes in those bacteriophages that were observed to be associated with CD medications in human patients. While a majority of bacteriophages were not influenced by medications, we found that a number of bacteriophages were impacted by 5-ASA and MP (increased Behunavirus BH1 by 5-ASA, **Supplementary figure 10B**; decreased Brevibacillus phage Sundance and Lactobacillus phage Lj928 by MP, **Supplementary figure 10C&F**), exhibiting an abundance change in the same directions seen in the medication-virome associations in patients. This data suggest that CD medications can causally change the abundances of a defined set of bacteriophages yet a lot of bacteriophage species may not vary as a

function of medication use. To further substantiate the impact of these medications on intestinal bacteriophages under physiological conditions, we conducted an in vivo experiment via administering 5-ASA, AZA, and MP respectively to microbiome-humanised mice, and then investigated the abundance change in the target gut bacteriophage post medication administration (**Supplementary figure 11**). Again, while the majority of the bacteriophages remained unchanged by these medications, a subset of bacteriophages were impacted by 5-ASA and MP, coinciding with the findings in our in vitro experiment (decreased *Lactobacillus* phage LJ928 with MP use) and in our human observations (decreased *Clostridium* phage PhiS63 with 5-ASA use, and decreased Coetzeevirus JL1 phage with MP use) (**Supplementary figures 11B&F, H**). Overall, these data together substantiated that medication use has a critical impact on the gut bacteriophage composition.

5-The DSS-induced colitis is normally very strong with robust evidence of colonic ulcerations, that, in my opinion, are not appreciable in Figure 7. In this regard, I suggest better images and indicate the Disease Activity Index as a combination of the different parameters to be evaluated in experimental models of intestinal inflammation.

Response: We agree with the reviewer's comment and appreciate the kind suggestion. To better present our data, we supplemented an endoscopic image indicating colonic ulcerations and a Disease Activity Index (DAI) evaluation data in this revision (Data are shown in **Figure 7D** and **Supplementary figure 15**).

The new data presentation is as below:

In order to validate the role of mucosal virome in intestinal inflammation, we replicated another batch of animal experiments in DSS mice, and evaluated DAI. In line with the phenotype observed in previous animal batches, mice administered with CD ileal VLPs exhibited a significantly more severe intestinal inflammation than those administered non-CD ileal VLPs post DSS treatment, as demonstrated by a shortened colon length, an increased DAI index, a higher histological inflammation score (*Mann-Whitney* test, $p=0.0241$, $p=0.041$, $p=0.0324$, respectively, **Supplementary figure 15B-F**), and elevated mRNA expression levels of pro-inflammatory cytokines (including TNF- α , IL-6, IL-17, *Mann-Whitney* test, all $p<0.05$, **Supplementary figure 15G**). All these findings are incorporated into this revision in **lines 628-629**.

Reviewer #2 (Remarks to the Author):

The manuscript titled "The gut ileal mucosal virome is disturbed in patients with Crohn's disease and exacerbates intestinal inflammation in mice," submitted by Zhirui et al., provides a comprehensive description of the ileal tissue virome landscape from patients with Crohn's disease (CD) in clinical remission or flare-up, and analyzes virus-bacteria interactions. This analysis utilizes data from two geographically distinct cohorts and includes extensive metadata information. Some revisions are necessary before publication.

We appreciate the reviewer's commendation on our study. We performed additional analyses and animal experiments to address the concerns kindly raised by the reviewer as below.

Major Points:

1. Given that tissue virome data was utilized, it is assumed that most of the data belong to the host. Could the authors quantify the proportion of virome data? This would aid in evaluating the quality of the virome data.

Response: We agree with the reviewer's comment. In most of the current human virome studies, contaminations from human reads and bacteria reads are a prevalent issue. Hence, in our study, all sequencing data was cleaned and decontaminated by removing human/bacterial host. Ultimately, 3.01% of the assembled, non-redundant contigs were identified as viral contigs, which is on par with other studies on faecal virome^{2,3}. Meanwhile, it is expected that given the mucosa/tissue-sourced nature of our data in the present study focusing on mucosal biome, a majority of sequencing reads were non-viral (and instead were human- and bacteria- derived). This information was added in **lines 118-119** in the **Supplementary materials**.

[2] Gregory AC, Zablocki O, Zayed AA, Howell A, Bolduc B, Sullivan MB. The Gut Virome Database Reveals Age-Dependent Patterns of Virome Diversity in the Human Gut. *Cell Host & Microbe* 28, 724-740.e728 (2020).

[3] Clooney AG, et al. Whole-Virome Analysis Sheds Light on Viral Dark Matter in Inflammatory Bowel Disease. *Cell Host Microbe* 26, 764-778 e765 (2019).

2. The abundance of lytic phages is notably higher than that of temperate phages in this study, contrary to previous understanding in stool (R. Sausset *et al.*, 2020). This difference may be attributed to the methodology employed to predict the phages' lifestyles. Or is it because of the ileal environment? Can the authors provide a more detailed discussion on this?

Response: We appreciate the reviewer's comment. After revisiting the existing literature on stool virome, including the one the reviewer kindly referred to (R. Sausset *et al.*, 2020), we found that temperate phages constituted approximately 20% to 50% of free phages in the human faeces^{4,5}. In our study which instead focused on mucosal phageome, we reported that temperate phages constituted 21.61% (median ratio) of the total phages on the mucosa level, which fallen within the reported range on the faeces level, despite at the lower end of the range. Overall, the abundance of lytic phages is notably higher than that

of temperate phages at the intestinal mucosa. Studies have shown that lytic phages could attach to the mucosal layer of the intestine through its Ig-like domains on the capsid, thereby guarding against invasions of pathogenic bacteria and hence maintaining epithelial homeostasis ^{6,7}. In addition, the ileal environment is different from that in the lumen and faeces, where the phages are kept in check simultaneously by mammalian epithelial cells, host bacteria, and available stress molecules. With all these factors at play, the mucosal phageome are anticipated to be different from the faecal phageome compositionally. This information is added into the **Discussion section** in **lines 780-790**.

[4] Sausset R, Petit MA, Gaboriau-Routhiau V, De Paepe M. New insights into intestinal phages. *Mucosal Immunology* 13, 205-215 (2020).

[5] Shah SA, et al. Expanding known viral diversity in the healthy infant gut. *Nature Microbiology* 8, 986-998 (2023).

[6] Barr JJ, et al. Bacteriophage adhering to mucus provide a non-host-derived immunity. *Proceedings of the National Academy of Sciences* 110, 10771-10776 (2013).

[7] Barr JJ, et al. Subdiffusive motion of bacteriophage in mucosal surfaces increases the frequency of bacterial encounters. *Proceedings of the National Academy of Sciences* 112, 13675-13680 (2015).

3. Figure 1E is unnecessary, as it has been reported many years ago.

Response: We have moved this figure down to Supplementary figure 2E.

4. The authors assert that geography has a greater impact on the bacterial community compared to inflammation. Further information is needed regarding the bacterial community. In patients with IBD, the expansion of Proteobacteria has been widely reported. What major differences exist between Guangzhou and Kunming?

Response: Thank you for your insightful comment. We performed additional analysis to discern the bacteria associated with geography (Guangzhou vs. Kunming) and CD. The major findings are shown in **lines 307-314** and in figures **Supplementary figure 7A&B, D** in this revision. The results are also briefed as below:

We identified the ileal mucosal bacteria associated with geography and CD respectively, by *MaAsLin2* analysis. Overall, geography had a larger effect size than CD in influencing the mucosal bacterial community (**Supplementary figure 7D**). At the phylum level, subjects in the Kunming cohort were enriched for Actinobacteriota, whilst subjects in the Guangzhou cohort were enriched for Bacteroidota, Desulfobacterota, Firmicutes, Fusobacteriota, and Proteobacteria, displaying a crucial geographic effect (**Supplementary figure 7D**). When comparing the ileal mucosal bacteriome between CD patients and HC, we found a significant expansion of Proteobacteria and a reduction of Firmicutes, in both the Guangzhou and Kunming cohorts (**Supplementary figure 7A&B, D**). However, CD patients in the Guangzhou cohort exhibited an additional depletion of Bacteroidota and Desulfobacterota, which were not observed in the Kunming cohort (**Supplementary figure 7A&B, D**). These data suggest a geographic effect in shaping the CD mucosal bacteriome across different populations.

5. Line 243: The Shannon diversity of the ileal virome was not significantly reduced in the CD flare-up group compared to the HC group in the Guangzhou cohort (Fig 2D).

Response: We apologize for not explaining our data accurately in our previous submission. We have now revised the sentence to read "The richness of the ileal virome was significantly decreased in the CD flare-up group than the HC group in both the Guangzhou and Kunming cohorts, while the Shannon diversity was significantly decreased in the CD flare-up group versus the HC group in the Kunming cohort (*Mann-Whitney* test, all $p < 0.01$, **Figure 2D&E**)." (shown in **lines 251-255**).

6. Line 279: Please clarify the percentage of virome data that can be assigned at the species level.

Response: We have added a message as per the reviewer's suggestion, clarifying that 47.8% of the assembled viral contigs were assigned at the species level (it is now shown in **lines 826-827**).

7. Line 424: Why is the abundance of a single bacterium correlated with numerous phage species with a similar correlation coefficient? Could the authors reevaluate the raw data? Please refer to a classic publication (Norman 2015).

Response: We appreciate the reviewer's comment. The reasons as to why we detected correlations between a single bacterium with multiple phage species are as below:

1) In the co-existing gut ecology of bacteria and phages, it is common to observe that a single phage may evolve to infect one or a few bacterial species⁸. Beyond that, given the large number of bacteria present in the human gut, a single bacterium can correlate with a number of other bacteria^{9,10}, and be infected simultaneously by several phages as well^{11,12}. Such biological behaviours and ecological observation may result in indirect correlations between a bacterium of interest and multiple phage species whose host bacteria also correlated with the bacterium of interest, from an ecology perspective.

2) In this study, to evaluate the correlations between bacteria and phages in this study, we adopted a recently customised analysis pipeline *Fastspar*¹³, a parallelizable C++ implementation of the *SparCC* algorithm¹⁴, which considered the diversity, sparsity, and density natures of the human microbiome data, so as to avoid missed detection of microbe-microbe associations due to low abundance or low prevalence of microbial species¹⁵. In contrast, the method of *Spearman* correlation analysis, used by the Norman et al. study (2015), was reported to be methodologically unfavourable in detecting associations between microbial species with low abundances or rare occurrences. But still, their result also found that a single bacterium correlated with a number of phage species with a similar correlation coefficient. We anticipate that *Fastspar* is more appropriate for analysing the sparse microbiome data, particularly considering that most of the gut phages are present in very low abundances. This is evidenced in the below figure, where we compared the *Fastspar* and *Spearman* analyses in evaluating the correlations between bacteria and phages in our data. We found that, while *Spearman* analysis detected a limited number of

phage-bacterium associations, *Fastspat* analysis detected a larger number of phage-bacterium associations.

[8] Ross A, Ward S, Hyman P. More Is Better: Selecting for Broad Host Range Bacteriophages. *Frontiers in Microbiology* 7, (2016).

[9] Chen L, et al. Gut microbial co-abundance networks show specificity in inflammatory bowel disease and obesity. *Nature Communications* 11, 4018 (2020).

[10] Steele JA, et al. Marine bacterial, archaeal and protistan association networks reveal ecological linkages. *The ISME Journal* 5, 1414-1425 (2011).

[11] Federici S, et al. Targeted suppression of human IBD-associated gut microbiota commensals by phage consortia for treatment of intestinal inflammation. *Cell* 185, 2879-2898. e2824 (2022).

[12] Díaz-Muñoz SL, Koskella B. Bacteria-phage interactions in natural environments. *Adv Appl Microbiol* 89, 135-183 (2014).

[13] Watts SC, Ritchie SC, Inouye M, Holt KE. FastSpar: rapid and scalable correlation estimation for compositional data. *Bioinformatics* 35, 1064-1066 (2018).

[14] Friedman J, Alm EJ. Inferring correlation networks from genomic survey data. *PLoS computational biology* 8, e1002687 (2012).

[15] Weiss S, et al. Correlation detection strategies in microbial data sets vary widely in sensitivity and precision. *Isme j* 10, 1669-1681 (2016).

8. Line 578: A significant point is the absence of experiments on DSS-induced mice without

any VLP treatment. This omission hinders a deeper understanding of the data.

Response: We appreciate this insightful comment. To address this concern, we conducted another batch of animal study and added a placebo group of mice which were only administered SM buffer, as a virome-absent control group. We found that the placebo group of mice exhibited insignificant inflammation phenotypes (colon length, DAI, proinflammatory cytokine expression) with the mice administered non-CD VLPs, while CD VLPs administration evoked the most significant inflammation phenotype in mice (**Supplementary figure 15B-G**). These data suggest that CD mucosal VLPs play a causal role in worsening intestinal inflammation in IBD mice. These new data are shown in the newly added figures **Supplementary figure 15** in the manuscript.

9. The discussion presented requires further elaboration. It is advisable to delve into previous publications that encompass both stool and tissue studies on IBD, in order to contextualize and enhance the depth of the current findings. Please refer to some publications (Wagner 2013; Norman 2015; Clooney 2019; Liang 2020,2021;).

Response: We are thankful for the reviewer's insightful comments. We added a paragraph to discuss our findings in the context of previous publications that encompassed both stool and tissue studies on IBD. They are in **lines 708-725**, highlighted in red and also shown as below:

CD is an autoimmune disease characterized by intermittent episodes of intestinal mucosal inflammation, whereby gut bacteriome dysbiosis was reported to contribute greatly to the pathogenesis of CD. Apart from bacteriome, there are an ample number of viruses in the gut contributing to defending against pathogens' invasion, modulating the bacteriome ecology, and regulating the mucosal immunity of the human host. Disturbance in the mucosal virome is herein postulated to underline CD onset and/or disease course. Most of the existing gut virome studies centered on the faecal virome. In faecal virome studies^{16, 17}, both CD and UC patients exhibited increased virome richness and diversity, characterized by an expansion of *Caudovirales* and a reduction in *Microviridae* abundance, compared to HC. These findings were also observed in the feces and intestinal tissues of paediatric CD patients based on a modest sample sized cohort^{18, 19, 20}. In addition, temperate phages were found to predominate the faecal virome in CD compared to HC³. However, to date, the mucosal virome, consisting of both phageome and eukaryotic virome, remains unclear in both health and CD, particularly on the small bowel level. Our study for the first time explored the composition and function of the ileal virome (at the small bowel mucosa level) in healthy adults and CD, in association with various clinical factors, including medications, diet, and geography.

[3] Clooney AG, et al. Whole-Virome Analysis Sheds Light on Viral Dark Matter in Inflammatory Bowel Disease. *Cell Host Microbe* 26, 764-778 e765 (2019)

[16] Norman JM, et al. Disease-specific alterations in the enteric virome in inflammatory bowel disease. *Cell* 160, 447-460 (2015).

[17] Kong C, Liu G, Kalady MF, Jin T, Ma Y. Dysbiosis of the stool DNA and RNA virome in Crohn's disease. *J Med Virol*, (2023).

[18] Liang G, et al. Dynamics of the Stool Virome in Very Early-Onset Inflammatory Bowel Disease. *J Crohns Colitis* 14, 1600-1610 (2020).

[19] Wagner J, et al. Bacteriophages in Gut Samples From Pediatric Crohn's Disease Patients: Metagenomic Analysis Using 454 Pyrosequencing. *Inflammatory Bowel Diseases* 19, 1598-1608 (2013).

[20] Liang G, Cobián-Güemes AG, Albenberg L, Bushman F. The gut virome in inflammatory bowel diseases. *Current Opinion in Virology* 51, 190-198 (2021).

Minor Points:

Line 205: Mislabeled? Figure 1G&I/1IJ?

Response: We have checked and corrected the figure numbering in Figure 1G&I. (shown in **line 199**)

Line 396: host -> hosts

Response: We have changed the word "host" to "hosts" (shown in **line 405**).

Line 541: should be "associated with"

Response: We have changed the phrase to "associated with" (shown in **line 577**).

Numerous grammar errors or typos, especially in the Methods section, require revision. Please carefully edit the manuscript.

Response: We appreciate the reviewer's constructive suggestions. We have thoroughly reviewed our manuscript to correct grammar errors or typos.

Newly added References:

1. Argelaguet R, et al. Multi-Omics Factor Analysis-a framework for unsupervised integration of multi-omics data sets. *Mol Syst Biol* 14, e8124 (2018).
2. Gregory AC, Zablocki O, Zayed AA, Howell A, Bolduc B, Sullivan MB. The Gut Virome Database Reveals Age-Dependent Patterns of Virome Diversity in the Human Gut. *Cell Host & Microbe* 28, 724-740.e728 (2020).
3. Clooney AG, et al. Whole-Virome Analysis Sheds Light on Viral Dark Matter in Inflammatory Bowel Disease. *Cell Host Microbe* 26, 764-778 e765 (2019).
4. Sausset R, Petit MA, Gaboriau-Routhiau V, De Paepe M. New insights into intestinal phages. *Mucosal Immunology* 13, 205-215 (2020).
5. Shah SA, et al. Expanding known viral diversity in the healthy infant gut. *Nature Microbiology* 8, 986-998 (2023).
6. Barr JJ, et al. Bacteriophage adhering to mucus provide a non-host-derived immunity. *Proceedings of the National Academy of Sciences* 110, 10771-10776 (2013).
7. Barr JJ, et al. Subdiffusive motion of bacteriophage in mucosal surfaces increases the frequency of bacterial encounters. *Proceedings of the National Academy of Sciences* 112, 13675-13680 (2015).
8. Ross A, Ward S, Hyman P. More Is Better: Selecting for Broad Host Range Bacteriophages. *Frontiers in Microbiology* 7, (2016).
9. Chen L, et al. Gut microbial co-abundance networks show specificity in inflammatory bowel disease and obesity. *Nature Communications* 11, 4018 (2020).
10. Steele JA, et al. Marine bacterial, archaeal and protistan association networks reveal ecological linkages. *The ISME Journal* 5, 1414-1425 (2011).
11. Federici S, et al. Targeted suppression of human IBD-associated gut microbiota commensals by phage consortia for treatment of intestinal inflammation. *Cell* 185, 2879-2898. e2824 (2022).
12. Díaz-Muñoz SL, Koskella B. Bacteria-phage interactions in natural environments. *Adv Appl Microbiol* 89, 135-183 (2014).
13. Watts SC, Ritchie SC, Inouye M, Holt KE. FastSpar: rapid and scalable correlation estimation for compositional data. *Bioinformatics* 35, 1064-1066 (2018).

14. Friedman J, Alm EJ. Inferring correlation networks from genomic survey data. *PLoS computational biology* 8, e1002687 (2012).
15. Weiss S, et al. Correlation detection strategies in microbial data sets vary widely in sensitivity and precision. *ISME J* 10, 1669-1681 (2016).
16. Norman JM, et al. Disease-specific alterations in the enteric virome in inflammatory bowel disease. *Cell* 160, 447-460 (2015).
17. Kong C, Liu G, Kalady MF, Jin T, Ma Y. Dysbiosis of the stool DNA and RNA virome in Crohn's disease. *J Med Virol*, (2023).
18. Liang G, et al. Dynamics of the Stool Virome in Very Early-Onset Inflammatory Bowel Disease. *J Crohns Colitis* 14, 1600-1610 (2020).
19. Wagner J, et al. Bacteriophages in Gut Samples From Pediatric Crohn's Disease Patients: Metagenomic Analysis Using 454 Pyrosequencing. *Inflammatory Bowel Diseases* 19, 1598-1608 (2013).
20. Liang G, Cobián-Güemes AG, Albenberg L, Bushman F. The gut virome in inflammatory bowel diseases. *Current Opinion in Virology* 51, 190-198 (2021).

REVIEWERS' COMMENTS

Reviewer #1 (Remarks to the Author):

I appreciate that the authors elegantly resolved all my issues. Moreover, I think this paper may add a very large piece of science to the IBD field.

Reviewer #2 (Remarks to the Author):

All of my comments have been appropriately addressed.